# A LINEAR ALGEBRAIC FRAMEWORK FOR COUNTERFACTUAL GENERATION

**Jong-Hoon Ahn & Akshay Vashist**
Otsuka Pharmaceutical Development & Commercialization, Inc.
Princeton, NJ 08540, USA
`{jong-hoon.ahn, akshay.vashist}@otsuka-us.com`

## ABSTRACT

Estimating individual treatment effects in clinical data is essential for understanding how different patients uniquely respond to treatments and identifying the most effective interventions for specific patient subgroups, thereby enhancing the precision and personalization of healthcare. However, counterfactual data are not accessible, and the true calculation of causal effects cannot be performed at the individual level. This paper proposes a linear algebraic framework to generate counterfactual longitudinal data that exactly matches pre-treatment factual data. Because causation travels forward in time, not in reverse, counterfactual predictability is further strengthened by blocking causal effects from flowing back to the past, thus limiting counterfactual dependence on the future. Using simulated LDL cholesterol datasets, we show that our method significantly outperforms the most cited methods of counterfactual generation. We also provide a formula that can estimate the time-varying variance of individual treatment effects, interpreted as a confidence level in the generated counterfactuals compared to true values.

## 1 INTRODUCTION

Estimating individual treatment effects (ITE) within clinical data is pivotal in advancing healthcare towards more personalized and effective treatment strategies (Simon & Perlis, 2010; Kent et al., 2018; Hoogland et al., 2021; Berchialla et al., 2022). It allows a nuanced understanding of how diverse patient populations uniquely respond to various treatments. This detailed insight is crucial for pinpointing the most beneficial interventions for specific patient subgroups, thereby significantly enhancing the precision and personalization of medical care. By tailoring treatments to patients' individual characteristics and conditions, healthcare providers can improve treatment outcomes, minimize adverse effects, and ultimately elevate patient satisfaction and quality of life.

However, a significant challenge in this endeavor is the inaccessibility of counterfactual data—information on what would have happened to a patient had a different treatment been administered. Without this data, accurately calculating the true causal effects of treatments at the individual level remains an infeasible task (Holland, 1986). This limitation hinders our ability to make definitive statements about the efficacy of one treatment over another for a specific patient. The absence of counterfactual scenarios in clinical data necessitates the development of sophisticated statistical models and methodologies (Shalit et al., 2017; Lu et al., 2018; Yoon et al., 2018) that can estimate these effects as closely as possible, using the available factual data and making informed assumptions about the unseen counterfactuals.

This paper proposes a generative model that can predict counterfactual data given observed longitudinal data, including multiple pre-treatment measurements. The use of multiple pre-treatment measurements in randomized clinical trials has been proposed in recent years (Van Patten et al., 2002; Vickers, 2003; Ma & Wang, 2023). Pre-treatment data are measured multiple times before participants enter a clinical trial for several key reasons (Friedman et al., 2015). Firstly, collecting data at various points can help establish a more accurate baseline of the participant's health status. By averaging these measurements, researchers can mitigate the effects of anomalies or fluctuations in individual health metrics, providing a clearer picture of the participant's condition before the intervention. Secondly, multiple pre-treatment measurements allow for the assessment of the stability or progression of a

condition over time, offering valuable insights into the natural history of the disease being studied. This information is crucial for interpreting the effectiveness of the treatment, as it helps to distinguish between changes caused by the treatment and those that might have occurred naturally. Lastly, but more importantly for our approach, these repeated measurements before intervention can improve the precision of the study's outcomes. By using these baseline values as covariates in analytical models, such as ANCOVA (Frison & Pocock, 1992), researchers can control for individual differences at the outset, thereby increasing the study's statistical power. This approach helps make more accurate inferences about the treatment's effect, ensuring that the observed changes are indeed attributable to the treatment rather than to pre-existing trends or variability in participant's conditions.

There have been efforts to use multiple pre-treatment measurements and their inherent temporal structure for counterfactual predictions (Abadie & Gardeazabal, 2003; Xu, 2017; Amjad et al., 2018; Qian et al., 2021; Doudchenko et al., 2021; Shao et al., 2022). However, most do not find counterfactual data whose pre-treatment data exactly matches the pre-treatment factual data. This paper introduces a linear algebraic framework to create counterfactual longitudinal data that aligns perfectly with observed pre-treatment factual data. The essence of this approach lies in its acknowledgment of the fundamental principle that causation operates in a forward temporal direction, not backward. This framework significantly enhances the predictability and reliability of counterfactual outcomes by preventing causal effects from retroactively altering past data, thus limiting the influence of future events on counterfactual scenarios. This methodological advance helps accurately model outcomes under various treatment scenarios, enhancing causal relationship insights. Such a framework is valuable not only in understanding clinical data but also in any field where understanding the precise impact of interventions over time is critical, enabling a more informed approach to decision-making based on the potential future effects of actions taken in the present.

## 2 METHODS

### 2.1 LEARNING STATIC-STATE REPRESENTATION FROM OBSERVATIONAL DATA

Our framework of counterfactual generation starts with simple assumptions.

**Assumption 1** (State). A real-world unit subject to an action, treatment, or regime is represented by a static *state* $\mathbf{s} \in \mathbb{R}^M$.

Examples of a real-world unit may include a physical object, a firm, a patient, an individual person, or a collection of objects or persons, such as a classroom or a market. The same unit exposed to an alternative action, treatment, or regime has a different static state. For our purposes, the same physical object or person at a different time remains the same unit.[1] Instead, a unit is observed in a time-varying form by the next assumption.

**Assumption 2** (Observation). A real-world unit represented by a static state $\mathbf{s}$ can be observed by a time-varying *observer* $\mathbf{W}^{(t)}$,

$$\mathbf{x}^{(t)} = \mathbf{W}^{(t)}\mathbf{s} + \boldsymbol{\eta}^{(t)} \tag{1}$$

where $\boldsymbol{\eta}^{(t)}$ is a measurement noise.

This state-space representation model may fall under the specific category[2] (explicit discrete time-variant systems) of the general state-space representation of a linear system in the control theory (Brogan, 1991).

The measurements made by linear observers are referred to as observational data.

**Definition 2.1** (Observational Data). $\mathbf{x}^{(t)} \in \mathbb{R}^D$ is the *observational data* of $\mathbf{s}$ at time $t$. $D$ is a sum of the number of outcomes of interest and the number of time-independent or -dependent covariates.

Suppose that we have $N$ units' longitudinal data, $\mathbb{D} = \{\mathbf{x}_{n,\omega_n}^{(t)} \in \mathbb{R}^D | \omega_n \in \{0,1\}; t = 1, \cdots, T; n = 1, \cdots, N\}$, where $\mathbf{x}_{n,\omega_n}$ is observed data of the $n$-th unit who received a treatment $\omega_n \in \{0,1\}$. The number of units that received $\omega_n = 0$ is $N_0$ and the number of units that

---

[1]It was assumed that the same object or person at a different time is a different unit by Imbens & Rubin (2015); Yao et al. (2021).

[2]Specifically, the system matrix $\mathbf{A}^{(t)}$, the input matrix $\mathbf{B}^{(t)}$, and the feedthrough matrix $\mathbf{D}^{(t)}$ are zero while the output matrix $\mathbf{C}^{(t)}$ is $\mathbf{W}^{(t)}$.

received $\omega_n = 1$ is $N_1$, which sum to $N = N_0 + N_1$. $\omega_n = 0$ or $\omega_n = 1$ can mean that the $n$-th unit belongs to a control or treatment group, respectively. Generally, each treatment may involve a fixed combination of multiple medications. For example, $\omega_n = 0$ or $\omega_n = 1$ with three different medications of $\{A, B, C\}$ and $T = 5$ can mean that the $n$-th patient takes $ABCBA$ or $ABABC$, respectively, where the first different medication is found at $t = 3$ and then we say the intervention time [3] is $T_0 = 3$. If the $n$-th patient took treatment $\omega_n = 0$ meaning $ABCBA$ and we obtained its factual outcomes, what would be probable outcomes if the patient had taken treatment $\omega_n = 1$ meaning $ABABC$? We can regard the first $T_0$ outcomes as the same to the observational data, but it is challenging to predict the next outcomes. The binary treatment cases are an example of the counterfactual generation problems that we try to solve, and it is straightforward to extend it to three or more treatments cases.

Assumption 2 raises a representational learning problem of how to find static states from those $N$ units' observational data by solving a system of $T$ equations given by Eq. (1). In this paper, we propose to use an $\omega$-conditional Gaussian mixture model (GMM) as a prior probability distribution over the static states by

$$p(\mathbf{s}|\omega) = \sum_{k=1}^{K_\omega} \pi_{k,\omega} \mathcal{N}(\mathbf{s}|\boldsymbol{\mu}_{k,\omega}, \boldsymbol{\Sigma}_{k,\omega}) \tag{2}$$

where $\pi_{k,\omega}$, $\boldsymbol{\mu}_{k,\omega}$, and $\boldsymbol{\Sigma}_{k,\omega}$ are the mixing proportion, mean vector, and covariance matrix of the $k$-th mixture component, respectively, for $\omega \in \{0, 1\}$. Eq. (1) implies a probability distribution over $\mathbf{x}^{(t)}$-space for a given $\mathbf{s}$ of the form

$$p(\mathbf{x}^{(t)}|\mathbf{s}) = (2\pi)^{-\frac{D}{2}} |\boldsymbol{\Psi}_o|^{-\frac{1}{2}} \exp\left\{-\frac{1}{2}(\mathbf{x}^{(t)} - \mathbf{W}^{(t)}\mathbf{s})^T \boldsymbol{\Psi}_o^{-1}(\mathbf{x}^{(t)} - \mathbf{W}^{(t)}\mathbf{s})\right\} \tag{3}$$

where $\boldsymbol{\Psi}_o \in \mathbb{R}^{D \times D}$ is a diagonal noise covariance matrix such that $\boldsymbol{\eta} \sim \mathcal{N}(\mathbf{0}, \boldsymbol{\Psi}_o)$. We can estimate those parameters of $\mathbb{Q} = \{\pi_{k,\omega}, \boldsymbol{\mu}_{k,\omega}, \boldsymbol{\Sigma}_{k,\omega}, \mathbf{W}^{(t)}, \boldsymbol{\Psi}_o | \omega \in \{0, 1\}; t = 1, \cdots, T; k = 1, \cdots, K_\omega\}$ and static states $\mathbf{s}$ by the EM algorithms for maximization of the complete data log-likelihood

$$\mathcal{L}_C = \sum_{n=1}^{N} \sum_{t=1}^{T} \ln p(\mathbf{x}_{n,\omega_n}^{(t)}, \mathbf{s}_{n,\omega_n}). \tag{4}$$

where the $n$-th unit's state $\mathbf{s}_{n,\omega_n}$ is regarded as missing data and given as an expectation $\langle \mathbf{s}_{n,\omega_n} \rangle$ by the EM algorithms derived in Appendix A. We refer to solving Eq. (1) by using the EM algorithms as a *static state analysis with priors of $\omega$-conditional Gaussian mixture model* or SSA-GMM.

Equation (2) is replaceable by any other model as long as it can universally approximate the underlying data distributions, but in this paper we use a GMM as a universal approximator of the state densities.

**Assumption 3** (Universal Approximator). A Gaussian mixture model is a universal approximator of densities, in the sense that any smooth density can be approximated with any specific nonzero amount of error by a Gaussian mixture model with enough components (Goodfellow et al., 2016).

## 2.2 Generating Counterfactual Data

Counterfactual thinking is imagining how things could have been different if something had changed in the past or future. If a different action, treatment, or regime had changed the state $\mathbf{s}$ of a unit by Assumption 1, the observational data would have been different through the observation of state $\mathbf{s}$ by Assumption 2. To get the counterfactual data, we need to imagine that the unit had been observed under the same-world conditions from which the observational data were obtained.

**Assumption 4** (Counterfactual Thinking in State-Space Model). When a unit received treatment $\omega$, we obtained its observational data $\mathbf{x}_\omega^{(t)}$ by observing its state $\mathbf{s}_\omega$ at time $t$, *i.e*,

$$\mathbf{x}_\omega^{(t)} = \mathbf{W}^{(t)}\mathbf{s}_\omega + \boldsymbol{\eta}^{(t)}.$$

If the unit had received an alternative treatment $\bar{\omega}$, we would have obtained its observational data $\mathbf{x}_{\bar{\omega}}^{(t)}$ through the same observational process for its state $\mathbf{s}_{\bar{\omega}}$, *i.e.*,

$$\mathbf{x}_{\bar{\omega}}^{(t)} = \mathbf{W}^{(t)}\mathbf{s}_{\bar{\omega}} + \boldsymbol{\eta}^{(t)},$$

which is a counterfactual conditional statement for the unit.

---

[3] The same notation, $T_0$, for intervention time was used in Abadie et al. (2010); Abadie (2021).

Since effects do not precede their causes, any counterfactual data observed from different treatments at time $T_0$ cannot differ from the factual data prior to $T_0$. We should ensure the independence of an intervention made at $T_0$ by using the first $T_0$ observers. Given a sequence of the first $T_0$ observers $\{\mathbf{W}^{(t)} \in \mathbb{R}^{D \times M} | D < M, 1 \le t \le T_0\}$, we let $\mathbf{N}^{(T_0)} \in \mathbb{R}^{M \times J}$ be an orthonormal basis for the null space such that $\mathbf{W}^{(t)}\mathbf{N}^{(T_0)} = \mathbf{O}$ for $t \le T_0$. To ensure no backward causation (Lewis, 1979), *i.e.*, $\mathbf{x}_\omega^{(t)} = \mathbf{x}_{\bar{\omega}}^{(t)}$ for $t \le T_0$, we need $\mathbf{W}^{(t)}(\mathbf{s}_{\bar{\omega}} - \mathbf{s}_\omega) = \mathbf{0}$, which means that the difference of two states must lie in the null space spanned by $\mathbf{N}^{(T_0)}$.

**Definition 2.2** (Counterfactual State). Suppose that we generate $\mathbf{s}_{\bar{\omega}} \sim p(\mathbf{s}|\bar{\omega})$ from Equation (2) subject to

$$\mathbf{s}_{\bar{\omega}} - \mathbf{s}_\omega = \mathbf{N}^{(T_0)}\boldsymbol{\xi} \tag{5}$$

with an arbitrary vector $\boldsymbol{\xi} \in \mathbb{R}^J$. Then, we call $\mathbf{s}_{\bar{\omega}}$ a *counterfactual state* that the unit would have if it had received the alternative treatment $\bar{\omega}$.

By the nature of a counterfactual state defined by Definition 2.2, counterfactual state reasoning is a random process defined as a sequence of independent and $\bar{\omega}$-conditionally distributed random states $\mathbf{s}_{\bar{\omega}}$ of Definition 2.2. Generating counterfactual data under each treatment $\bar{\omega}$ are a deterministic process of the observation of each random state, as described in Assumption 4. To understand Eq. (5), let us recall:

**Definition 2.3** (Affine Subspace). A subset $\mathbb{U} \subset \mathbb{R}^M$ is called an affine subspace of $\mathbb{R}^M$ if all affine combinations of vectors in $\mathbb{U}$ remain also in $\mathbb{U}$:

$$\mathbf{u}_1, \cdots, \mathbf{u}_{J+1} \in \mathbb{U}, \; \lambda_1, \cdots, \lambda_{J+1} \in \mathbb{R} \text{ with } \sum_{j=1}^{J+1} \lambda_j = 1 \implies \sum_{j=1}^{J+1} \lambda_j \mathbf{u}_j \in \mathbb{U}.$$

If we put $\mathbf{u}_j \equiv [\mathbf{N}^{(T_0)}]_j + \mathbf{s}_\omega$ for $j = 1, \cdots, J$ and $\mathbf{u}_{J+1} \equiv \mathbf{s}_\omega$ with $\lambda_j \equiv \xi_j$ for $j = 1, \cdots, J$ and $\lambda_{J+1} \equiv 1 - \sum_{j=1}^J \xi_j$, we can find that $\mathbf{s}_{\bar{\omega}} \in \mathbb{U}$ from Eq. (5).

The use of a GMM in Eq. (2) enables us to directly generate a counterfactual state. For that, we need two lemmas:

**Lemma 2.1** (Conditional Distribution of a Multivariate Normal Distribution). *Given a multivariate normal distribution $\mathcal{N}(\mathbf{s}|\boldsymbol{\mu}, \boldsymbol{\Sigma})$ and an affine subspace $\mathbb{U}_0 = \{\mathbf{s} \mid \mathbf{s} - \mathbf{s}_0 = \mathbf{W}_0 \boldsymbol{\xi}, \; \boldsymbol{\xi} \in \mathbb{R}^J\}$ with $\mathbf{W}_0 \in \mathbb{R}^{M \times J}$ and $M > J$, the conditional probability density function $p(\boldsymbol{\xi}|\mathbf{s} \in \mathbb{U}_0)$ is also a normal distribution and is given by $\mathcal{N}(\boldsymbol{\xi}|\mathbf{m}, \mathbf{C})$ where the conditional mean and variance are*

$$\begin{aligned} \mathbf{m} &= (\mathbf{W}_0^T \boldsymbol{\Sigma}^{-1} \mathbf{W}_0)^{-1} \mathbf{W}_0^T \boldsymbol{\Sigma}^{-1}(\boldsymbol{\mu} - \mathbf{s}_0) \\ \mathbf{C} &= (\mathbf{W}_0^T \boldsymbol{\Sigma}^{-1} \mathbf{W}_0)^{-1}. \end{aligned}$$

**Lemma 2.2** (Conditional Distribution of a Mixture of Multivariate Normal Distributions). *Given a mixture of multivariate normal distributions $\sum_k \pi_k \mathcal{N}(\mathbf{s}|\boldsymbol{\mu}_k, \boldsymbol{\Sigma}_k)$ with $\sum_k \pi_k = 1$ and an affine subspace $\mathbb{U}_0 = \{\mathbf{s} \mid \mathbf{s} - \mathbf{s}_0 = \mathbf{W}_0 \boldsymbol{\xi}, \; \boldsymbol{\xi} \in \mathbb{R}^J\}$ with $\mathbf{W}_0 \in \mathbb{R}^{M \times J}$ and $M > J$, the conditional probability density function $p(\boldsymbol{\xi}|\mathbf{s} \in \mathbb{U}_0)$ is also a mixture of normal distributions and is given by $\sum_k p_k \mathcal{N}(\boldsymbol{\xi}|\mathbf{m}_k, \mathbf{C}_k)$ where the conditional proportions are*

$$p_k = \frac{\pi_k \mathcal{N}(\mathbf{W}_0 \mathbf{m}_k + \mathbf{s}_0 | \boldsymbol{\mu}_k, \boldsymbol{\Sigma}_k) |\mathbf{C}_k|^{\frac{1}{2}}}{\sum_{k'} \pi_{k'} \mathcal{N}(\mathbf{W}_0 \mathbf{m}_{k'} + \mathbf{s}_0 | \boldsymbol{\mu}_{k'}, \boldsymbol{\Sigma}_{k'}) |\mathbf{C}_{k'}|^{\frac{1}{2}}}$$

*with*

$$\begin{aligned} \mathbf{m}_k &= (\mathbf{W}_0^T \boldsymbol{\Sigma}_k^{-1} \mathbf{W}_0)^{-1} \mathbf{W}_0^T \boldsymbol{\Sigma}_k^{-1}(\boldsymbol{\mu}_k - \mathbf{s}_0) \\ \mathbf{C}_k &= (\mathbf{W}_0^T \boldsymbol{\Sigma}_k^{-1} \mathbf{W}_0)^{-1}. \end{aligned}$$

Finally, we make our main contribution to counterfactual generation:

**Theorem 2.3** (Gasussian Mixture Counterfactual Generator). *Let $\mathbb{Q}$ be a maximizer of Equation (4) for a factual dataset $\mathbb{D}$. Adding $\mathbf{W}^{(t)}\mathbf{N}^{(T_0)}\boldsymbol{\xi}_{n,\bar{\omega}_n}$ to $\mathbf{x}_{n,\omega_n}^{(t)}$ generates counterfactual data of the $n$-th unit,*

$$\mathbf{x}_{n,\bar{\omega}_n}^{(t)} = \mathbf{x}_{n,\omega_n}^{(t)} + \mathbf{W}^{(t)}\mathbf{N}^{(T_0)}\boldsymbol{\xi}_{n,\bar{\omega}_n} \tag{6}$$

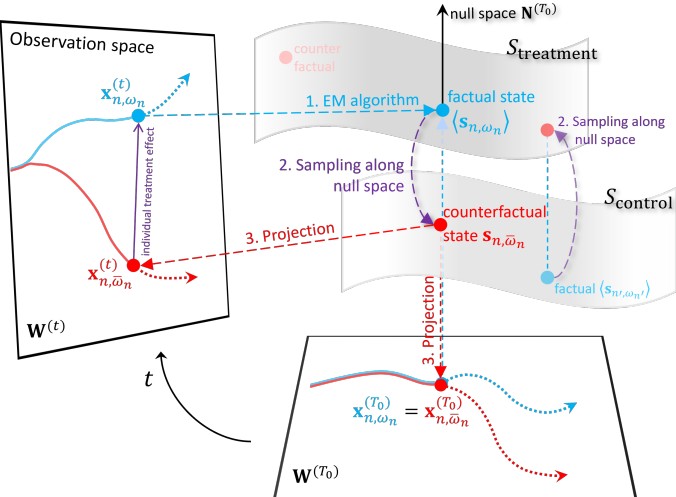

Figure 1: A conceptual view of how synthetic counterfactual longitudinal data are generated.

*where* $\boldsymbol{\xi}_{n,\bar{\omega}_n}$ *is a random vector such that*

$$\boldsymbol{\xi}_{n,\bar{\omega}_n} \sim \sum_{k=1}^{K_{\bar{\omega}_n}} p_{k,\bar{\omega}_n} \mathcal{N}(\boldsymbol{\xi}|\mathbf{m}_{k,\bar{\omega}_n}, \mathbf{C}_{k,\bar{\omega}_n}) \tag{7}$$

*with*

$$\mathbf{C}_{k,\bar{\omega}_n} = (\mathbf{N}^{(T_0)T}\boldsymbol{\Sigma}_{k,\bar{\omega}_n}^{-1}\mathbf{N}^{(T_0)})^{-1} \tag{8}$$

$$\mathbf{m}_{k,\bar{\omega}_n} = \mathbf{C}_{k,\bar{\omega}_n}\mathbf{N}^{(T_0)T}\boldsymbol{\Sigma}_{k,\bar{\omega}_n}^{-1}(\boldsymbol{\mu}_{k,\bar{\omega}_n} - \langle\mathbf{s}_{n,\omega_n}\rangle) \tag{9}$$

$$p_{k,\bar{\omega}_n} = \frac{\pi_{k,\bar{\omega}_n}\mathcal{N}(\mathbf{N}^{(T_0)}\mathbf{m}_{k,\bar{\omega}_n} + \langle\mathbf{s}_{n,\omega_n}\rangle|\boldsymbol{\mu}_{k,\bar{\omega}_n}, \boldsymbol{\Sigma}_{k,\bar{\omega}_n})|\mathbf{C}_{k,\bar{\omega}_n}|^{\frac{1}{2}}}{\sum_{k'=1}^{K_{\bar{\omega}_n}} \pi_{k',\bar{\omega}_n}\mathcal{N}(\mathbf{N}^{(T_0)}\mathbf{m}_{k',\bar{\omega}_n} + \langle\mathbf{s}_{n,\omega_n}\rangle|\boldsymbol{\mu}_{k',\bar{\omega}_n}, \boldsymbol{\Sigma}_{k',\bar{\omega}_n})|\mathbf{C}_{k',\bar{\omega}_n}|^{\frac{1}{2}}}. \tag{10}$$

**Corollary 2.3.1** (No Backward Causation). *Any counterfactual data* $\mathbf{x}_{n,\bar{\omega}}^{(t)}$ *do not deviate from the factual data* $\mathbf{x}_{n,\omega}^{(t)}$ *until the alteration of treatments at* $t = T_0$.

Figure 1 shows a conceptual summary view of counterfactual generation. Curved data manifolds denoted by $S_{control}$ and $S_{treat}$ are estimated as $p(\mathbf{s}|\omega = 0)$ and $p(\mathbf{s}|\omega = 1)$ of Eq. (2), respectively, by maximizing the log-likelihood function of Eq. (4) in terms of GMM parameters – proportions $\pi_{k,\omega}$, means $\boldsymbol{\mu}_{k,\omega}$, and covariances $\boldsymbol{\Sigma}_{k,\omega}$. We can generate synthetic factual data using their own GMM parameters, which is not of primary interest in this paper. For a counterfactual generation, from the (blue) factual point on $S_{treat}$, for example, we need to sample out a new (red) point $\mathbf{s}_{n,\omega_n}$ from a GMM of the opposite manifold $S_{control}$, which is annotated with "sampling along null space" and performed by using Eq. (5). Eq. (5) includes a random variable $\boldsymbol{\xi}$, sampled from the conditional distribution, Eq. (7). After randomly generating a counterfactual state $\mathbf{s}_{n,\bar{\omega}_n}$, we can finally obtain the time-varying counterfactual data by projecting the fixed point $\mathbf{s}_{n,\bar{\omega}_n}$ onto the moving lower-dimensional observational space spanned by $\mathbf{W}^{(t)}$ or, equivalently, using Eq. (1). Here, any generated counterfactual data are equal to the observed factual data at $t \leq T_0$, which is stated in Corollary 2.3.1. After $t = T_0$, it may deviate from the factual data.

## 2.3 ESTIMATING INDIVIDUALIZED TREATMENT EFFECT

In order to estimate an ITE, we need to read out the outcomes-only difference because $\mathbf{x}_{n,\omega_n}^{(t)} \in \mathbb{R}^D$ includes covariates as well as outcomes of interest for the $n$-th unit at time $t$. The individualized treatment effect (ITE) for the $n$-th subject after the treatment time $t = T_0$ with $\omega_n \in \{0, 1\}$ is defined as:

$$\text{ITE}_n^{(t)} = (1 - 2\omega_n)\mathbf{w}_{\text{effect}}^T(\mathbf{x}_{n,\bar{\omega}_n}^{(t)} - \mathbf{x}_{n,\omega_n}^{(t)}) \tag{11}$$

where $\mathbf{x}_{n,\bar{\omega}_n}^{(t)}$ is a synthetic counterfactual data of the $n$-th subject's factual data, $\mathbf{x}_{n,\omega_n}^{(t)}$. $\mathbf{w}_{\text{effect}}$ is a column vector to return an ITE selecting one of the differences between factual and counterfactual data. For example, it could be $[1, 0, \cdots, 0]^T$ if the first component of $\mathbf{x}_n^{(t)}$ is the outcome of interest.

**Corollary 2.3.2** (Individualized Treatment Effect). Let $\mathbb{Q}$ be a maximizer of Equation (4) for a factual dataset $\mathbb{D}$. The individualized treatment effect through an effect operator $\mathbf{w}_{\text{effect}}$ is given by

$$\text{ITE}_n^{(t)} = (1 - 2\omega_n)\mathbf{w}_{\text{effect}}^T\mathbf{W}^{(t)}\mathbf{N}^{(T_0)}\boldsymbol{\xi}_{n,\bar{\omega}_n} \tag{12}$$

where $\boldsymbol{\xi}_{n,\bar{\omega}_n}$ is given by Eq. (7).

**Remark 1.** Equation (12) can be read as

$$\text{ITE}^{(t)} = \mathbf{w}_{\text{effect}}^T \cdot \mathbf{W}^{(t)} \cdot (1 - 2\omega)\mathbf{N}^{(T_0)}\boldsymbol{\xi} = (\text{readout}) \circ (\text{observer}) \circ (\text{treatment effect}).$$

Equation (12) gives a random value. We can also calculate its mean and variance.

**Theorem 2.4** (Expectation-Variance of ITE). *Let $\mathbb{Q}$ be a maximizer of Equation (4) for a factual dataset $\mathbb{D}$, the expected value and variance of individualized treatment effects for the $n$-th unit are given by*

$$\text{E}[\text{ITE}_n^{(t)}] = (1 - 2\omega_n)\mathbf{w}_{\text{effect}}^T\mathbf{W}^{(t)}\mathbf{N}^{(T_0)}\left(\sum_{k=1}^{K_{\omega_n}} p_{k,\bar{\omega}_n}\mathbf{m}_{k,\bar{\omega}_n}\right) \tag{13}$$

$$\text{Var}[\text{ITE}_n^{(t)}] = \mathbf{w}_{\text{effect}}^T\mathbf{W}^{(t)}\mathbf{N}^{(T_0)}\left(\sum_{k=1}^{K_{\omega_n}} p_{k,\bar{\omega}_n}\mathbf{C}_{k,\bar{\omega}_n}^{\text{total}}\right)\mathbf{N}^{(T_0)T}\mathbf{W}^{(t)T}\mathbf{w}_{\text{effect}} \tag{14}$$

*where*

$$\mathbf{C}_{k,\bar{\omega}_n}^{\text{total}} = \mathbf{C}_{k,\bar{\omega}_n}^{\text{within}} + \mathbf{C}_{k,\bar{\omega}_n}^{\text{between}} \equiv \mathbf{C}_{k,\bar{\omega}_n} + (\mathbf{m}_{k,\bar{\omega}_n} - \overline{\mathbf{m}}_{\bar{\omega}_n})(\mathbf{m}_{k,\bar{\omega}_n} - \overline{\mathbf{m}}_{\bar{\omega}_n})^T$$

$$\overline{\mathbf{m}}_{\bar{\omega}_n} = \sum_{k'} p_{k',\bar{\omega}_n}\mathbf{m}_{k',\bar{\omega}_n}.$$

## 3 EXPERIMENTS

### 3.1 CALIFORNIA CIGARETTE SALES DATA

We demonstrate Corollary 2.3.1 and Theorem 2.4 by using a public example: the problem of estimating the effect of cigarette taxation on per-capita cigarette consumption. In 1988, California introduced the first modern-time large-scale anti-tobacco legislation in the United States. To analyze the effect of California's legislation after 1988, we use the annual per-capita cigarette consumption at the state level for 38 states not adopting anti-tobacco legislation programs between 1970 and 2000.

The data set has $T = 31$ (years of observation), $N = N_0 + N_1 = 38$ (untreated states) + 1 (California), and $D = 1$. For analysis, we set $(M, K_0)$ to be $(30, 2)$ and trained our SSA-GMM method for only control group data of 38 untreated states. The expectation curve of the synthetic counterfactual data of California is estimated by using Eq. (13) and is plotted in Fig. 2. We can observe that the pretreatment synthetic California data exactly matches the factual California data between 1970 and 1988, as described in Corollary 2.3.1. Our method is a random generation, and the standard deviations are plotted with pink filling, which is computed using Eq. (14).

The mean counterfactual curve comes from the convex combination of $K_0$ centers, denoted by $\mathbf{m}_{\bar{\omega}k}$, with nonnegative and sum-to-one weights, denoted by $p_{\bar{\omega}k}$, as described in Eq. (13). Interestingly, using a convex combination of nonnegative and sum-to-one weights is also the original key idea of the synthetic control method (Abadie & Gardeazabal, 2003). Accurate pretreatment reconstruction by the synthetic control method (SC) or any SC-based method usually gives an overfitted post-treatment result. In contrast, our post-treatment result shows no overfitting despite the pretreatment fitting and a convex combination.

There are slight differences between our post-treatment prediction and the results of the synthetic control method after 1994. Which is more true cannot be determined; it agrees with the Bayesian

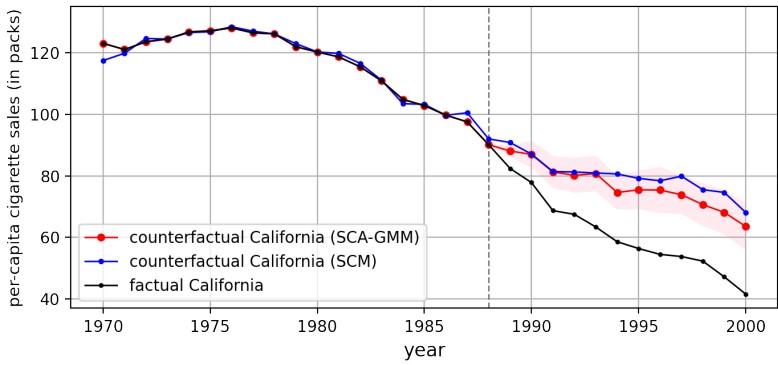

Figure 2: Application of SSA-GMM to California cigarette consumption data.

approach result of Amjad et al. (2018) showing that it may be possible that the synthetic control method overestimated the effect of California's legislation. Evaluating how much true synthetic counterfactuals are in ITE estimation requires a simulated dataset, including ground truth counterfactual data, as in the following section.

## 3.2 SIMULATED LDL CHOLESTEROL DATA

To evaluate our counterfactual generation method provided by Theorem 2.3, we used the simulated LDL cholesterol dataset that had been used to evaluate the SyncTwin algorithm (Qian et al., 2021). We aimed to estimate the LDL cholesterol-lowering effect of statins, a drug commonly prescribed to hypercholesterolaemic patients. The dataset is created using the widely adopted pharmacology model in the literature (Faltaos et al., 2006; Kim et al., 2011). For more information about the simulation study, refer to the paper of Qian et al. (2021).

Table 1: Mean absolute errors (MAE) on ITE under different levels of confounding bias $p_0$ and data size $N_0$. It is assumed that every unit received treatment at $T_0 = 25$ and there are no missing data. Estimated standard deviations are shown in parentheses. The best performer is in bold.

| Method | $N = N_0 + N_1 = 200 + 200 = 400$ | | | $N = N_0 + N_1 = 1000 + 200 = 1200$ | | |
|---|---|---|---|---|---|---|
| | $p_0 = 0.1$ | $p_0 = 0.25$ | $p_0 = 0.5$ | $p_0 = 0.1$ | $p_0 = 0.25$ | $p_0 = 0.5$ |
| SSA-GMM | **0.072**(.004) | **0.072**(.004) | **0.076**(.004) | **0.070**(.004) | **0.073**(.004) | **0.069**(.004) |
| SyncTwin | 0.308 (.037) | 0.150 (.013) | 0.116 (.008) | 0.178 (.012) | 0.106 (.006) | 0.094 (.005) |
| SC | 0.341 (.042) | 0.151 (.024) | 0.150 (.018) | 0.231 (.033) | 0.172 (.031) | 0.158 (.023) |
| RSC | 0.846 (.046) | 0.357 (.021) | 0.299 (.018) | 0.296 (.016) | 0.284 (.014) | 0.292 (.015) |
| MC-NNM | 1.160 (.060) | 0.613 (.032) | 0.226 (.012) | 0.526 (.029) | 0.177 (.010) | 0.124 (.006) |
| CFRNet | 0.888 (.076) | 0.466 (.043) | 0.279 (.027) | 0.236 (.018) | 0.139 (.008) | 0.113 (.007) |
| CRN | 0.530 (.035) | 0.631 (.048) | 0.343 (.023) | 0.456 (.027) | 0.404 (.028) | 0.371 (.030) |
| RMSN | 0.646 (.043) | 0.608 (.040) | 0.538 (.038) | 0.720 (.053) | 0.635 (.046) | 0.601 (.043) |
| CGP | 0.674 (.043) | 0.619 (.039) | 0.552 (.035) | 0.825 (.055) | 0.690 (.046) | 0.602 (.038) |

To explain how and why our algorithm is robust to bias, we analyzed an outlier sample, which is shown as the black dashed line in Fig. 3, in the test dataset of $\omega = 1$. The dashed curve is represented by the yellow point of $\langle \mathbf{s}_n \rangle$. There are no similar curves in the $\omega = 0$ group test data, and the In our notation, the pre-treatment time steps go from $t = 1$ through $t = 25$, where $T_0 = 25$, and the post-treatment effects are estimated at $t = 26$ through $t = 30$. The dataset includes two covariates and one outcome, which yields $D = 3$. The big difference between our approach and the SyncTwin method is that we also use the post-treatment data of covariates and pre-treatment data of outcomes. In contrast, SyncTwin uses only pre-treatment data of covariates and post-treatment data of outcomes. To produce Table 1 as a reproduction of Table 2 of the SyncTwin paper (Qian et al., 2021), we trained our model with $K_0 = K_1 = 2$ and $M = 85$ from a dataset of $N_0 = N_1 = 200$ and a dataset of $N_0 = 1000$ and $N_1 = 200$. We also introduced confounding bias denoted by $p_n$ for the $n$-th patient:

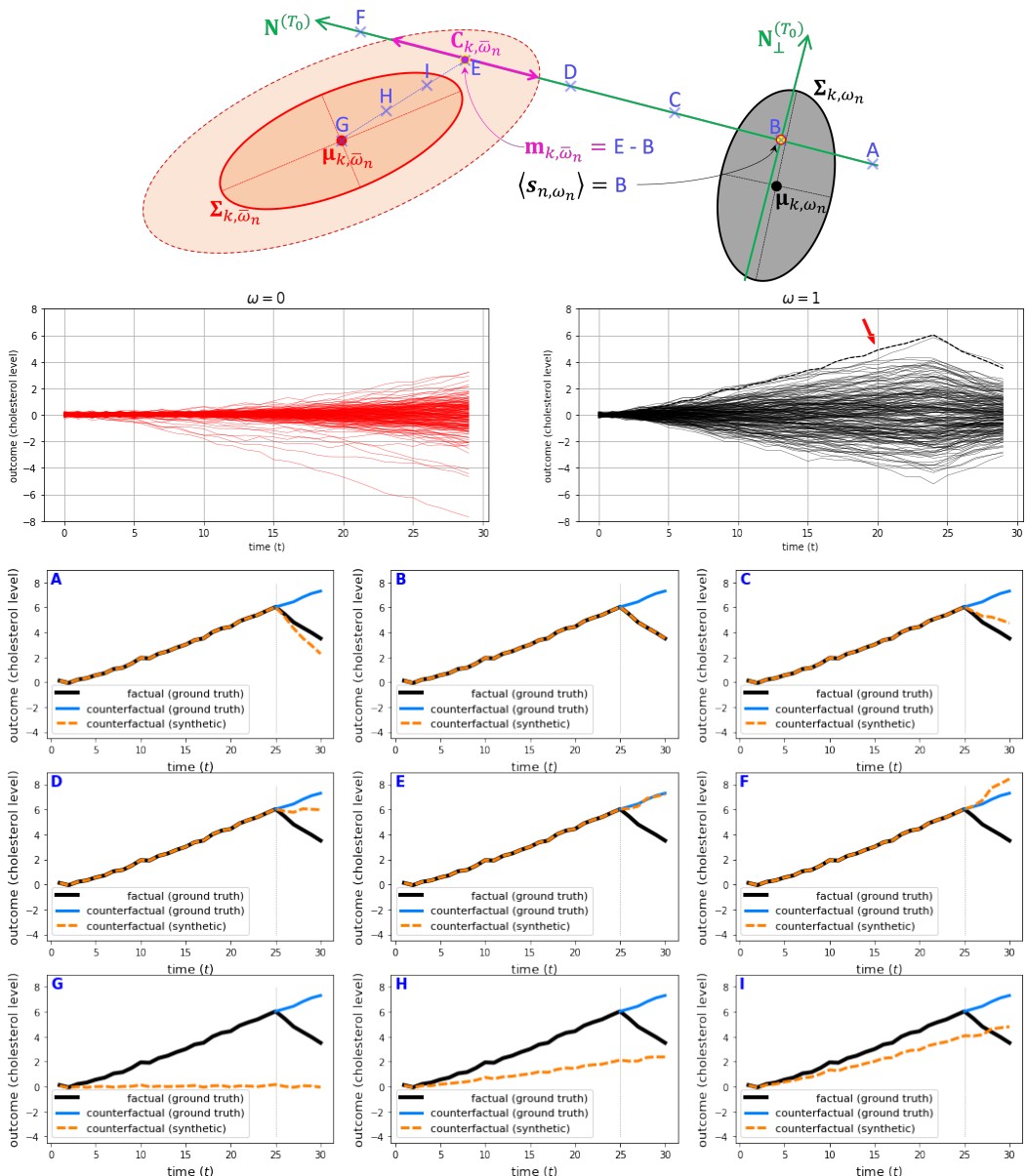

Figure 3: Outlier sample analysis. (upper) A conceptual view of the $k$-th Gaussian component and sampling points *A-I*; (center) Test outcome datasets of $\omega = 0$ group and $\omega = 1$ group; (lower) Outcome curves generated by SSA-GMM from the points.

$p_n = p_0$ for $\omega_n = 0$ and $p_n = 1$ for $\omega_n = 1$. The constant $p_0$ controls the degree of confounding bias (smaller $p_0$, larger bias).

Table 1 shows our method's improved performance compared to the methods of SC (Abadie et al., 2010), RSC (Amjad et al., 2018), MC-NNM (Athey et al., 2021), CFRNet (as a SyncTwin-combined form of Shalit et al. (2017)), CRN (Bica et al., 2020), RMSN (Lim, 2018), CGP (Schulam & Saria, 2017), and SyncTwin. Our model is especially robust to confounding bias $p_0$. As exemplified in supplementary Fig. 4, counterfactual pretreatment curves generated randomly by Eq. (6) and Eq. (7) exactly match their pretreatment factual data. Moreover, those generated counterfactual curves are not susceptible to post-treatment overfitting. A family of synthetic control methods directly exploring

a convex combination of real controls or transformed controls (e.g., Equation (4) of the SyncTwin paper) does not achieve this property.

To explain how and why our algorithm is robust to bias, we analyzed an outlier sample, which is shown as the black dashed line in Fig. 3, in the test dataset of $\omega = 1$. The dashed curve is represented by the yellow point of $\langle \mathbf{s}_n \rangle$. There are no similar curves in the $\omega = 0$ group test data, and the probability density along the null space is extremely thin. In our model, it is given by $\mathcal{N}(\langle \mathbf{s}_n \rangle + \mathbf{N}^{(T_0)} \boldsymbol{\xi} | \boldsymbol{\mu}_{k,\bar{\omega}_n}, \boldsymbol{\Sigma}_{k,\bar{\omega}_n})$ for any $\boldsymbol{\xi}$, which is extremely thin. Most algorithms listed in Table 1 fail to generate a counterfactual to this outlier. However, the *cut and normalized* (or, equivalently, conditional) probability distribution $\mathcal{N}(\boldsymbol{\xi} | \mathbf{m}_{k,\bar{\omega}_n}, \mathbf{C}_{k,\bar{\omega}_n})$, as given by Lemma 2.2 and cut by the null space (pink arrow), can generate a good counterfactual (see the subplot E). Generally, cutting a probability density function by the null space and normalizing the cross-sectional density function make it robust to bias. For comparison, we generated curves on the in-sample points of *A-F* over the null space and the out-sample points of *G-I*.

Table 2: Mean absolute errors by SSA-GMM on ITE under different noise levels $\sigma$. The second row is the same as the first row of Table 1.

| Method | $N = N_0 + N_1 = 200 + 200 = 400$ | | | $N = N_0 + N_1 = 1000 + 200 = 1200$ | | |
|---|---|---|---|---|---|---|
| | $p_0 = 0.1$ | $p_0 = 0.25$ | $p_0 = 0.5$ | $p_0 = 0.1$ | $p_0 = 0.25$ | $p_0 = 0.5$ |
| $\sigma = 0.01$ | 0.029 (.001) | 0.019 (.001) | 0.097 (.002) | 0.026 (.001) | 0.016 (.001) | 0.014 (.001) |
| $\sigma = 0.1$ | 0.072 (.004) | 0.072 (.004) | 0.076 (.004) | 0.070 (.004) | 0.073 (.004) | 0.069 (.004) |
| $\sigma = 0.2$ | 0.183 (.011) | 0.144 (.008) | 0.146 (.008) | 0.177 (.010) | 0.143 (.008) | 0.139 (.008) |

The dataset used for producing Table 1 was created by adding additional noise $\mathcal{N}(0, \sigma = 0.1)$ to a dataset made by solving the PK/PD model. Table 2 shows results by different noise effects with $\sigma = 0.01$ and $\sigma = 0.2$. Our algorithm's MAE (0.183 and 0.144) at $\sigma = 0.2$ are better than SyncTwin's MAE (0.308 and 0.150) at $\sigma = 0.1$, which strongly suggests that our method is robust to additional noise.

## 4  CONCLUSION

We have introduced the static state analysis with a Gaussian mixture model (SSA-GMM) and its use as a generative model for synthetic counterfactual longitudinal data. We have also demonstrated its ability to generate a counterfactual of individual-level observed data. Importantly, it can estimate the individual treatment effects (ITE) that change only after treatment using the synthetic counterfactual outcomes data or the formulas of expectation and variance of ITE. Using LDL cholesterol datasets that we simulated from a PK/PD model, we have shown that our method is robust to confounding bias and noise, greatly outperforming the most cited methods of counterfactual generation and ITE estimation.

ACKNOWLEDGEMENTS

We would like to thank Hossain Saboonchi and Anton Palma for their useful discussions and feedback on the early draft of this work, and Ksenia Korshunova for helping experimental work.

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

# A DERIVATION OF EXPECTATION-MAXIMIZATION ALGORITHMS

## A.1 STATIC STATE ANALYSIS WITH A GAUSSIAN MIXTURE MODEL

For the case of having noise $\boldsymbol{\eta} \sim \mathcal{N}(\mathbf{0}, \boldsymbol{\Psi}_{\mathrm{o}})$ with a diagonal matrix $\boldsymbol{\Psi}_{\mathrm{o}} \in \mathbb{R}^{D \times D}$, Eq. (1) implies a probability distribution over $\mathbf{x}^{(t)}$-space for a given $\mathbf{s} \in \mathbb{R}^M$ of the form

$$p(\mathbf{x}^{(t)}|\mathbf{s}) = (2\pi)^{-\frac{D}{2}} |\boldsymbol{\Psi}_{\mathrm{o}}|^{-\frac{1}{2}} \exp\left\{ -\frac{1}{2}(\mathbf{x}^{(t)} - \mathbf{W}^{(t)}\mathbf{s})^T \boldsymbol{\Psi}_{\mathrm{o}}^{-1}(\mathbf{x}^{(t)} - \mathbf{W}^{(t)}\mathbf{s}) \right\}. \tag{15}$$

With a Gaussian mixture prior over $\mathbf{s}$ defined by

$$p(\mathbf{s}) = \sum_{k=1}^{K} \pi_k \mathcal{N}(\mathbf{s}|\boldsymbol{\mu}_k, \boldsymbol{\Sigma}_k), \tag{16}$$

we obtain the marginal distribution of $\mathbf{x}^{(t)}$ in the form

$$p(\mathbf{x}^{(t)}) = \sum_{k=1}^{K} \pi_k \cdot |2\pi \mathbf{C}_k^{(t)}|^{-\frac{1}{2}} \exp\left\{ -\frac{1}{2}(\mathbf{x}^{(t)} - \mathbf{W}^{(t)}\boldsymbol{\mu}_k)^T \mathbf{C}_k^{(t)}{}^{-1}(\mathbf{x}^{(t)} - \mathbf{W}^{(t)}\boldsymbol{\mu}_k) \right\}, \tag{17}$$

where the model covariance is $\mathbf{C}_k^{(t)} = \boldsymbol{\Psi}_{\mathrm{o}} + \mathbf{W}^{(t)}\boldsymbol{\Sigma}_k \mathbf{W}^{(t)T}$.

However, the joint distribution of all the observations $\{\mathbf{x}^{(1)}, \cdots, \mathbf{x}^{(T)}\}$ is intractable with

$$\begin{aligned}
p(\mathbf{x}^{(1)}, \cdots, \mathbf{x}^{(T)}) &= p(\mathbf{x}^{(1)}, \cdots, \mathbf{x}^{(T-1)}|\mathbf{x}^{(T)})p(\mathbf{x}^{(T)}) \\
&= p(\mathbf{x}^{(1)}, \cdots, \mathbf{x}^{(T-2)}|\mathbf{x}^{(T-1)}, \mathbf{x}^{(T)})p(\mathbf{x}^{(T-1)}|\mathbf{x}^{(T)})p(\mathbf{x}^{(T)}) \\
&\ \ \vdots
\end{aligned}$$

Let us define $\mathbf{x}$ and $\mathbf{W}$ by

$$\mathbf{x} = \left[ (\mathbf{x}^{(1)})^T (\mathbf{x}^{(2)})^T \cdots (\mathbf{x}^{(T)})^T \right]^T \in \mathbb{R}^{TD} \tag{18}$$

$$\mathbf{W} = \left[ (\mathbf{W}^{(1)})^T (\mathbf{W}^{(2)})^T \cdots (\mathbf{W}^{(T)})^T \right]^T \in \mathbb{R}^{(TD) \times M}. \tag{19}$$

Then, we obtain the conditional distribution and the marginal distribution in the form of

$$p(\mathbf{x}|\mathbf{s}) = |2\pi \boldsymbol{\Psi}|^{-\frac{1}{2}} \exp\left\{ -\frac{1}{2}(\mathbf{x} - \mathbf{W}\mathbf{s})^T \boldsymbol{\Psi}^{-1}(\mathbf{x} - \mathbf{W}\mathbf{s}) \right\} \tag{20}$$

$$p(\mathbf{x}) = \sum_k p(\mathbf{x}|k)p(k)$$

$$= \sum_k \pi_k \cdot |2\pi \mathbf{C}_k|^{-\frac{1}{2}} \exp\left\{ -\frac{1}{2}(\mathbf{x} - \mathbf{W}\boldsymbol{\mu}_k)^T \mathbf{C}_k^{-1}(\mathbf{x} - \mathbf{W}\boldsymbol{\mu}_k) \right\}. \tag{21}$$

where $\mathbf{C}_k = \boldsymbol{\Psi} + \mathbf{W}\boldsymbol{\Sigma}_k \mathbf{W}^T$ and $\boldsymbol{\Psi} = \mathbf{I}_{T \times T} \otimes \boldsymbol{\Psi}_{\mathrm{o}} \in \mathbb{R}^{(TD) \times (TD)}$ by the Kronecker product $\otimes$. By Bayes' rule, this leads to the posterior distribution of the form

$$\begin{aligned}
p(\mathbf{s}|\mathbf{x}) &= \sum_k p(\mathbf{s}|\mathbf{x}, k)p(k|\mathbf{x}) = \sum_k p(k|\mathbf{x}) \cdot p(\mathbf{x}|\mathbf{s})p(\mathbf{s}|k)/p(\mathbf{x}|k) \\
&= \sum_k p(k|\mathbf{x}) \cdot |2\pi \mathbf{M}_k|^{-\frac{1}{2}} \exp\Big\{ -\frac{1}{2}\big((\mathbf{s} - \boldsymbol{\mu}_k) - \mathbf{M}_k \mathbf{W}^T \boldsymbol{\Psi}^{-1}(\mathbf{x} - \mathbf{W}\boldsymbol{\mu}_k)\big)^T \\
&\qquad\qquad\qquad \times \mathbf{M}_k^{-1}\big((\mathbf{s} - \boldsymbol{\mu}_k) - \mathbf{M}_k \mathbf{W}^T \boldsymbol{\Psi}^{-1}(\mathbf{x} - \mathbf{W}\boldsymbol{\mu}_k)\big) \Big\}
\end{aligned} \tag{22}$$

where $\mathbf{M}_k = (\boldsymbol{\Sigma}_k^{-1} + \mathbf{W}^T \boldsymbol{\Psi}^{-1} \mathbf{W})^{-1}$.

Now suppose that we have observational data $\{\mathbf{x}_n^{(t)}\}$ for $n = 1, \cdots, N$ and $t = 1, \ldots, T$. We can maximize the log-likelihood function of Eq. (4) for the joint distribution of observational data $\mathbf{x}_n^{(t)}$ and the latent variables $\mathbf{s}_n$ in the form of

$$
\begin{aligned}
p(\mathbf{x}_n, \mathbf{s}_n) &= |2\pi\boldsymbol{\Psi}|^{-\frac{1}{2}} \exp\left\{-\frac{1}{2}(\mathbf{x}_n - \mathbf{W}\mathbf{s}_n)^T \boldsymbol{\Psi}^{-1}(\mathbf{x}_n - \mathbf{W}\mathbf{s}_n)\right\} \times \\
&\quad \sum_{k=1}^{K} \pi_k |2\pi\boldsymbol{\Sigma}_k|^{-\frac{1}{2}} \exp\left\{-\frac{1}{2}(\mathbf{s}_n - \boldsymbol{\mu}_k)^T \boldsymbol{\Sigma}_k^{-1}(\mathbf{s}_n - \boldsymbol{\mu}_k)\right\}.
\end{aligned}
\tag{23}
$$

By calculating the expectation values

$$
\gamma_{nk} \equiv p(k|\mathbf{x}_n) \tag{24}
$$

$$
\langle \mathbf{s}_n \rangle_k = \int p(\mathbf{s}|\mathbf{x}_n, k)\, \mathbf{s}\, |d\mathbf{s}| \tag{25}
$$

$$
\langle \mathbf{s}_n \rangle = \int p(\mathbf{s}|\mathbf{x}_n)\, \mathbf{s}\, |d\mathbf{s}| \tag{26}
$$

$$
\langle (\mathbf{s}_n - \langle \mathbf{s}_n \rangle)(\mathbf{s}_n - \langle \mathbf{s}_n \rangle)^T \rangle = \sum_k p(k|\mathbf{x}_n)\, \mathbf{M}_k, \tag{27}
$$

we can obtain the expectation of the complete data log-likelihood in the form of

$$
\begin{aligned}
\langle \mathcal{L}_C \rangle &= \sum_{n=1}^{N} \sum_{k=1}^{K} \gamma_{nk} \Big\{ \ln \pi_k - \frac{1}{2}\ln|\boldsymbol{\Sigma}_k| - \frac{1}{2}\mathrm{tr}(\boldsymbol{\Sigma}_k^{-1}\langle \mathbf{s}_n \mathbf{s}_n^T \rangle) + \boldsymbol{\mu}_k^T \boldsymbol{\Sigma}_k^{-1}\langle \mathbf{s}_n \rangle - \frac{1}{2}\boldsymbol{\mu}_k^T \boldsymbol{\Sigma}_k^{-1}\boldsymbol{\mu}_k \\
&\quad - \frac{1}{2}\ln|\boldsymbol{\Psi}| - \frac{1}{2}\mathbf{x}_n^T \boldsymbol{\Psi}^{-1}\mathbf{x}_n + \mathbf{x}_n^T \boldsymbol{\Psi}^{-1}\mathbf{W}\langle \mathbf{s}_n \rangle - \frac{1}{2}\mathrm{tr}(\mathbf{W}^T \boldsymbol{\Psi}^{-1}\mathbf{W}\langle \mathbf{s}_n \mathbf{s}_n^T \rangle) \Big\} \\
&\quad - \lambda(\sum_{k=1}^{K} \pi_k - 1)
\end{aligned}
\tag{28}
$$

where the expectation values are given by

E-steps:

$$
\gamma_{nk} = \frac{\pi_k \mathcal{N}(\mathbf{x}_n | \mathbf{W}\boldsymbol{\mu}_k, \mathbf{C}_k)}{\sum_{k'} \pi_{k'} \mathcal{N}(\mathbf{x}_n | \mathbf{W}\boldsymbol{\mu}_{k'}, \mathbf{C}_{k'})} \tag{29}
$$

$$
\langle \mathbf{s}_n \rangle_k = \boldsymbol{\mu}_k + \mathbf{M}_k \mathbf{W}^T \boldsymbol{\Psi}^{-1}(\mathbf{x}_n - \mathbf{W}\boldsymbol{\mu}_k) \tag{30}
$$

$$
\langle \mathbf{s}_n \rangle = \sum_k \gamma_{nk} \langle \mathbf{s}_n \rangle_k \tag{31}
$$

$$
\langle \mathbf{s}_n \mathbf{s}_n^T \rangle = \sum_k \gamma_{nk} \left( \mathbf{M}_k + (\langle \mathbf{s}_n \rangle_k - \langle \mathbf{s}_n \rangle)(\langle \mathbf{s}_n \rangle_k - \langle \mathbf{s}_n \rangle)^T \right) + \langle \mathbf{s}_n \rangle \langle \mathbf{s}_n \rangle^T \tag{32}
$$

with $\mathbf{C}_k = \boldsymbol{\Psi} + \mathbf{W}\boldsymbol{\Sigma}_k\mathbf{W}^T$ and $\mathbf{M}_k = (\boldsymbol{\Sigma}_k^{-1} + \mathbf{W}^T\boldsymbol{\Psi}^{-1}\mathbf{W})^{-1}$.

Equation (28) is maximized by the following M-step formulas

M-steps:

$$\pi_k = \frac{\sum_n \gamma_{nk}}{N} \tag{33}$$

$$\mu_k = \frac{1}{\sum_n \gamma_{nk}} \sum_n \gamma_{nk} \langle \mathbf{s}_n \rangle \tag{34}$$

$$\Sigma_k = \frac{1}{\sum_n \gamma_{nk}} \sum_n \gamma_{nk} (\langle \mathbf{s}_n \mathbf{s}_n^T \rangle - \mu_k \langle \mathbf{s}_n \rangle^T - \langle \mathbf{s}_n \rangle \mu_k^T + \mu_k \mu_k^T) \tag{35}$$

$$\mathbf{W}^{(t)} = \left[ \sum_n \mathbf{x}_n^{(t)} \langle \mathbf{s}_n \rangle^T \right] \left[ \sum_n \langle \mathbf{s}_n \mathbf{s}_n^T \rangle \right]^{-1} \tag{36}$$

$$\Psi_{\mathrm{o}} = \frac{1}{NT} \sum_{n=1}^N \mathrm{tr}_D \left[ \mathbf{x}_n \mathbf{x}_n^T - 2\mathbf{W} \langle \mathbf{s}_n \rangle \mathbf{x}_n^T + \mathbf{W} \langle \mathbf{s}_n \mathbf{s}_n^T \rangle \mathbf{W}^T \right] \tag{37}$$

where $\mathrm{tr}_D[\Psi]$ is a $d \times d$ diagonal matrix defined in this paper as returning a diagonal matrix

$$\mathrm{tr}_D \Psi \equiv \begin{pmatrix} \psi_{\mathrm{o},11} & 0 & 0 & 0 \\ 0 & \psi_{\mathrm{o},22} & 0 & 0 \\ 0 & 0 & \ddots & 0 \\ 0 & 0 & 0 & \psi_{\mathrm{o},DD} \end{pmatrix} \tag{38}$$

with $\psi_{\mathrm{o},dd} \equiv \sum_{t=0}^{T-1} \Psi_{(Dt+d)(Dt+d)}$. Notice that $\mathrm{tr}_1[\Psi]$ is equal to the trace of $\Psi$.

## A.2 STATIC STATE ANALYSIS WITH A CONDITIONAL GAUSSIAN MIXTURE MODEL

Now, we are ready to derive the $\omega$-conditional EM algorithms. Eq. (28) should be rewritten as

$$
\begin{aligned}
\langle \mathcal{L}_C \rangle =& \sum_{n=1}^N \sum_{k=1}^{K_{\omega_n}} \gamma_{nk} \Big\{ \ln \pi_{k,\omega_n} - \frac{1}{2} \ln |\Sigma_{k,\omega_n}| - \frac{1}{2} \mathrm{tr}(\Sigma_{k,\omega_n}^{-1} \langle \mathbf{s}_{n,\omega_n} \mathbf{s}_{n,\omega_n}^T \rangle) + \mu_{k,\omega_n}^T \Sigma_{k,\omega_n}^{-1} \langle \mathbf{s}_{n,\omega_n} \rangle \\
&- \frac{1}{2} \mu_{k,\omega_n}^T \Sigma_{k,\omega_n}^{-1} \mu_{k,\omega_n} - \frac{1}{2} \ln |\Psi| - \frac{1}{2} \mathbf{x}_{n,\omega_n}^T \Psi^{-1} \mathbf{x}_{n,\omega_n} + \mathbf{x}_{n,\omega_n}^T \Psi^{-1} \mathbf{W} \langle \mathbf{s}_{n,\omega_n} \rangle \\
&- \frac{1}{2} \mathrm{tr}(\mathbf{W}^T \Psi^{-1} \mathbf{W} \langle \mathbf{s}_{n,\omega_n} \mathbf{s}_{n,\omega_n}^T \rangle) \Big\} - \lambda \Big( \sum_{k=1}^{K_{\omega_n}} \pi_{k,\omega_n} - 1 \Big)
\end{aligned} \tag{39}
$$

where the E-steps are given as:

E-steps:

$$\gamma_{nk} = \frac{\pi_{k,\omega_n} \mathcal{N}(\mathbf{x}_{n,\omega_n} | \mathbf{W} \mu_{k,\omega_n}, \mathbf{C}_{k,\omega_n})}{\sum_{k'=1}^{K_{\omega_n}} \pi_{k',\omega_n} \mathcal{N}(\mathbf{x}_{n,\omega_n} | \mathbf{W} \mu_{k',\omega_n}, \mathbf{C}_{k',\omega_n})} \tag{40}$$

$$\langle \mathbf{s}_{n,\omega_n} \rangle_k = \mu_{k,\omega_n} + \mathbf{M}_{k,\omega_n} \mathbf{W}^T \Psi^{-1}(\mathbf{x}_{n,\omega_n} - \mathbf{W} \mu_{k,\omega_n}) \tag{41}$$

$$\langle \mathbf{s}_{n,\omega_n} \rangle = \sum_{k=1}^{K_{\omega_n}} \gamma_{nk} \langle \mathbf{s}_{n,\omega_n} \rangle_k \tag{42}$$

$$
\begin{aligned}
\langle \mathbf{s}_{n,\omega_n} \mathbf{s}_{n,\omega_n}^T \rangle =& \sum_{k=1}^{K_{\omega_n}} \gamma_{nk} \big( \mathbf{M}_{k,\omega_n} + (\langle \mathbf{s}_{n,\omega_n} \rangle_k - \langle \mathbf{s}_{n,\omega_n} \rangle)(\langle \mathbf{s}_{n,\omega_n} \rangle_k - \langle \mathbf{s}_{n,\omega_n} \rangle)^T \big) \\
&+ \langle \mathbf{s}_{n,\omega_n} \rangle \langle \mathbf{s}_{n,\omega_n} \rangle^T
\end{aligned} \tag{43}
$$

with $\mathbf{C}_{k,\omega} = \Psi + \mathbf{W} \Sigma_{k,\omega} \mathbf{W}^T$ and $\mathbf{M}_{k,\omega} = (\Sigma_{k,\omega}^{-1} + \mathbf{W}^T \Psi^{-1} \mathbf{W})^{-1}$ for $k = 1, \cdots, K_\omega$ with $\omega \in \{0, 1\}$.

Equation (39) can be maximized by

M-steps:

$$\pi_{k,\omega} = \frac{\sum_n \delta_{\omega\omega_n}\gamma_{nk}}{\sum_{n'}\delta_{\omega\omega_{n'}}} \tag{44}$$

$$\boldsymbol{\mu}_{k,\omega} = \frac{\sum_n \delta_{\omega\omega_n}\gamma_{nk}\langle\mathbf{s}_{n,\omega_n}\rangle}{\sum_{n'}\delta_{\omega\omega_{n'}}\gamma_{n'k}} \tag{45}$$

$$\boldsymbol{\Sigma}_{k,\omega} = \frac{\sum_n \delta_{\omega\omega_n}\gamma_{nk}(\langle\mathbf{s}_{n,\omega_n}\mathbf{s}_{n,\omega_n}^T\rangle - \boldsymbol{\mu}_{k,\omega}\langle\mathbf{s}_{n,\omega_n}\rangle^T - \langle\mathbf{s}_{n,\omega_n}\rangle\boldsymbol{\mu}_{k,\omega}^T + \boldsymbol{\mu}_{k,\omega}\boldsymbol{\mu}_{k,\omega}^T)}{\sum_{n'}\delta_{\omega\omega_{n'}}\gamma_{n'k}} \tag{46}$$

$$\mathbf{W} = \left[\sum_n \mathbf{x}_{n,\omega_n}\langle\mathbf{s}_{n,\omega_n}\rangle^T\right]\left[\sum_n\langle\mathbf{s}_{n,\omega_n}\mathbf{s}_{n,\omega_n}^T\rangle\right]^{-1} \tag{47}$$

$$\boldsymbol{\Psi}_{\mathrm{o}} = \frac{1}{NT}\sum_{n=1}^{N}\mathrm{tr}_D\left[\mathbf{x}_{n,\omega_n}\mathbf{x}_{n,\omega_n}^T - 2\mathbf{W}\langle\mathbf{s}_{n,\omega_n}\rangle\mathbf{x}_{n,\omega_n}^T + \mathbf{W}\langle\mathbf{s}_{n,\omega_n}\mathbf{s}_{n,\omega_n}^T\rangle\mathbf{W}^T\right] \tag{48}$$

where $\delta_{\omega\omega_n}$ is 1 for $\omega = \omega_n$ or 0 for $\omega \neq \omega_n$.

## A.3 STATIC STATE ANALYSIS WITH INCOMPLETE DATA

Suppose that $\mathbf{x}$ defined by Eq. (18) is incomplete data and we separate it into

$$\mathbf{x} = \boldsymbol{\Gamma}^{\mathrm{obs}}\mathbf{x}^{\mathrm{obs}} + \boldsymbol{\Gamma}^{\mathrm{mis}}\mathbf{x}^{\mathrm{mis}} \tag{49}$$

where $\mathbf{x}^{\mathrm{obs}}$ and $\mathbf{x}^{\mathrm{mis}}$ are observed data and missing data, respectively, with $\dim(\mathbf{x}^{\mathrm{obs}}) + \dim(\mathbf{x}^{\mathrm{mis}}) = TD$. We assume that $\boldsymbol{\Gamma}^{\mathrm{obs}}$ and $\boldsymbol{\Gamma}^{\mathrm{mis}}$ are constant matrices only having elements of 0 or 1 such that

$$\mathbf{x}^{\mathrm{obs}} = \boldsymbol{\Gamma}^{\mathrm{obs}T}\mathbf{x} \tag{50}$$

$$\mathbf{x}^{\mathrm{mis}} = \boldsymbol{\Gamma}^{\mathrm{mis}T}\mathbf{x} \tag{51}$$

or, equivalently, $\boldsymbol{\Gamma}^{\mathrm{obs}T}\boldsymbol{\Gamma}^{\mathrm{obs}} = \mathbf{I}$, $\boldsymbol{\Gamma}^{\mathrm{mis}T}\boldsymbol{\Gamma}^{\mathrm{mis}} = \mathbf{I}$, and $\boldsymbol{\Gamma}^{\mathrm{obs}T}\boldsymbol{\Gamma}^{\mathrm{mis}} = \mathbf{O}$.

The noise covariance matrix $\boldsymbol{\Psi}$ is diagonal, and the conditional distribution is separated as $p(\mathbf{x}|\mathbf{s}) = p(\mathbf{x}^{\mathrm{obs}}|\mathbf{s})p(\mathbf{x}^{\mathrm{mis}}|\mathbf{s})$ in the form of

$$p(\mathbf{x}^{\mathrm{obs}}|\mathbf{s}) = |2\pi\boldsymbol{\Psi}^{\mathrm{obs}}|^{-\frac{1}{2}}\exp\left\{-\frac{1}{2}(\mathbf{x}^{\mathrm{obs}} - \mathbf{W}^{\mathrm{obs}}\mathbf{s})^T\boldsymbol{\Psi}^{\mathrm{obs}-1}(\mathbf{x}^{\mathrm{obs}} - \mathbf{W}^{\mathrm{obs}}\mathbf{s})\right\} \tag{52}$$

$$p(\mathbf{x}^{\mathrm{mis}}|\mathbf{s}) = |2\pi\boldsymbol{\Psi}^{\mathrm{mis}}|^{-\frac{1}{2}}\exp\left\{-\frac{1}{2}(\mathbf{x}^{\mathrm{mis}} - \mathbf{W}^{\mathrm{mis}}\mathbf{s})^T\boldsymbol{\Psi}^{\mathrm{mis}-1}(\mathbf{x}^{\mathrm{mis}} - \mathbf{W}^{\mathrm{mis}}\mathbf{s})\right\} \tag{53}$$

where $\boldsymbol{\Psi}^{\mathrm{obs}} = \boldsymbol{\Gamma}^{\mathrm{obs}T}\boldsymbol{\Psi}\boldsymbol{\Gamma}^{\mathrm{obs}}$, $\boldsymbol{\Psi}^{\mathrm{mis}} = \boldsymbol{\Gamma}^{\mathrm{mis}T}\boldsymbol{\Psi}\boldsymbol{\Gamma}^{\mathrm{mis}}$, $\mathbf{W}^{\mathrm{obs}} = \boldsymbol{\Gamma}^{\mathrm{obs}T}\mathbf{W}$, and $\mathbf{W}^{\mathrm{mis}} = \boldsymbol{\Gamma}^{\mathrm{mis}T}\mathbf{W}$. With a Gaussian mixture prior over the latent variables defined by Eq. (16), we obtain the marginal distribution of observed data in the form of

$$p(\mathbf{x}^{\mathrm{obs}}) = \sum_{k=1}^{K}\pi_k \cdot |2\pi\mathbf{C}_k^{\mathrm{obs}}|^{-\frac{1}{2}}\exp\left\{-\frac{1}{2}(\mathbf{x}^{\mathrm{obs}} - \mathbf{W}^{\mathrm{obs}}\boldsymbol{\mu}_k)^T\mathbf{C}_k^{\mathrm{obs}-1}(\mathbf{x}^{\mathrm{obs}} - \mathbf{W}^{\mathrm{obs}}\boldsymbol{\mu}_k)\right\} \tag{54}$$

where $\mathbf{C}_k^{\mathrm{obs}} = \boldsymbol{\Psi}^{\mathrm{obj}} + \mathbf{W}^{\mathrm{obj}}\boldsymbol{\Sigma}_{k,\omega_n}\mathbf{W}^{\mathrm{obj}T}$. By Bayes' rule, this leads to the posterior distribution of the form

$$p(\mathbf{s}|\mathbf{x}^{\mathrm{obs}}) = \sum_k p(k|\mathbf{x}^{\mathrm{obs}}) \cdot |2\pi\mathbf{M}_k^{\mathrm{obs}}|^{-\frac{1}{2}}\exp\left\{-\frac{1}{2}\left(\mathbf{s} - \langle\mathbf{s}\rangle_k\right)^T\mathbf{M}_k^{\mathrm{obs}-1}\left(\mathbf{s} - \langle\mathbf{s}\rangle_k\right)\right\} \tag{55}$$

where

$$\mathbf{M}_k^{\mathrm{obs}} = (\boldsymbol{\Sigma}_k^{-1} + \mathbf{W}^{\mathrm{obs}T}\boldsymbol{\Psi}^{\mathrm{obs}-1}\mathbf{W}^{\mathrm{obs}})^{-1} \tag{56}$$

$$\langle\mathbf{s}\rangle_k = \boldsymbol{\mu}_k + \mathbf{M}_k^{\mathrm{obs}}\mathbf{W}^{\mathrm{obs}T}\boldsymbol{\Psi}^{\mathrm{obs}-1}(\mathbf{x}^{\mathrm{obs}} - \mathbf{W}^{\mathrm{obs}}\boldsymbol{\mu}_k). \tag{57}$$

Equation (28) is rewritten as

$$
\begin{aligned}
\langle \mathcal{L}_C \rangle \quad = & \sum_{n=1}^{N} \sum_{k=1}^{K_{\omega_n}} \{ \ln \pi_{k,\omega_n} - \frac{1}{2} \ln |\boldsymbol{\Sigma}_{k,\omega_n}| - \frac{1}{2} \mathrm{tr}(\boldsymbol{\Sigma}_{k,\omega_n}^{-1} \langle \mathbf{s}_{n,\omega_n} \mathbf{s}_{n,\omega_n}^T \rangle) + \boldsymbol{\mu}_{k,\omega_n}^T \boldsymbol{\Sigma}_{k,\omega_n}^{-1} \langle \mathbf{s}_{n,\omega_n} \rangle \\
& - \frac{1}{2} \boldsymbol{\mu}_{k,\omega_n}^T \boldsymbol{\Sigma}_{k,\omega_n}^{-1} \boldsymbol{\mu}_{k,\omega_n} - \frac{1}{2} \mathbf{x}_{n,\omega_n}^{\mathrm{obs}\,T} \boldsymbol{\Psi}_n^{\mathrm{obs}\,-1} \mathbf{x}_{n,\omega_n}^{\mathrm{obs}} + \mathbf{x}_{n,\omega_n}^{\mathrm{obs}\,T} \boldsymbol{\Psi}_n^{\mathrm{obs}\,-1} \mathbf{W}_n^{\mathrm{obs}} \langle \mathbf{s}_{n,\omega_n} \rangle \\
& - \frac{1}{2} \mathrm{tr}(\boldsymbol{\Psi}_n^{\mathrm{obs}\,-1} \mathbf{W}_n^{\mathrm{obs}} \langle \mathbf{s}_{n,\omega_n} \mathbf{s}_{n,\omega_n}^T \rangle \mathbf{W}_n^{\mathrm{obs}\,T}) - \frac{1}{2} \ln |\boldsymbol{\Psi}| \} \times \gamma_{nk} - \lambda(\sum_{k=1}^{K} \pi_{k,\omega_n} - 1) \quad (58)
\end{aligned}
$$

where $\mathbf{x}_{n,\omega_n}^{\mathrm{obs}} = \boldsymbol{\Gamma}_n^{\mathrm{obs}\,T} \mathbf{x}_{n,\omega_n}$, $\mathbf{W}_n^{\mathrm{obs}} = \boldsymbol{\Gamma}_n^{\mathrm{obs}\,T} \mathbf{W}$, $\boldsymbol{\Psi}_n^{\mathrm{obs}} = \boldsymbol{\Gamma}_n^{\mathrm{obs}\,T} \boldsymbol{\Psi} \boldsymbol{\Gamma}_n^{\mathrm{obs}}$, and $\boldsymbol{\Psi} = \mathbf{I} \otimes \boldsymbol{\Psi}_{\mathrm{o}}$.

Equation (58) is calculated by using

E-steps:

$$
\gamma_{nk} \quad = \quad \frac{\pi_{k,\omega_n} \mathcal{N}(\mathbf{x}_{n,\omega_n}^{\mathrm{obs}} | \mathbf{W}_n^{\mathrm{obs}} \boldsymbol{\mu}_{k,\omega_n}, \mathbf{C}_{k,\omega_n}^{\mathrm{obs}})}{\sum_{k'} \pi_{k',\omega_n} \mathcal{N}(\mathbf{x}_{n,\omega_n}^{\mathrm{obs}} | \mathbf{W}_n^{\mathrm{obs}} \boldsymbol{\mu}_{k',\omega_n}, \mathbf{C}_{k',\omega_n}^{\mathrm{obs}})} \quad (59)
$$

$$
\langle \mathbf{s}_{n,\omega_n} \rangle_k \quad = \quad \boldsymbol{\mu}_{k,\omega_n} + \mathbf{M}_{k,\omega_n}^{\mathrm{obs}} \mathbf{W}_n^{\mathrm{obs}\,T} \boldsymbol{\Psi}_n^{\mathrm{obs}\,-1} (\mathbf{x}_{n,\omega_n}^{\mathrm{obs}} - \mathbf{W}_n^{\mathrm{obs}} \boldsymbol{\mu}_{k,\omega_n}) \quad (60)
$$

$$
\langle \mathbf{s}_{n,\omega_n} \rangle \quad = \quad \sum_{k=1}^{K_{\omega_n}} \gamma_{nk} \langle \mathbf{s}_{n,\omega_n} \rangle_k \quad (61)
$$

$$
\begin{aligned}
\langle \mathbf{s}_{n,\omega_n} \mathbf{s}_{n,\omega_n}^T \rangle \quad = & \quad \sum_{k=1}^{K_{\omega_n}} \gamma_{nk} \left( \mathbf{M}_{k,\omega_n}^{\mathrm{obs}} + (\langle \mathbf{s}_{n,\omega_n} \rangle_k - \langle \mathbf{s}_{n,\omega_n} \rangle)(\langle \mathbf{s}_{n,\omega_n} \rangle_k - \langle \mathbf{s}_{n,\omega_n} \rangle)^T \right) \\
& + \langle \mathbf{s}_{n,\omega_n} \rangle \langle \mathbf{s}_{n,\omega_n} \rangle^T \quad (62)
\end{aligned}
$$

with $\mathbf{C}_{k,\omega_n}^{\mathrm{obs}} = \boldsymbol{\Psi}_n^{\mathrm{obj}} + \mathbf{W}_n^{\mathrm{obj}} \boldsymbol{\Sigma}_{k,\omega_n} \mathbf{W}_n^{\mathrm{obj}\,T}$ and $\mathbf{M}_{k,\omega_n}^{\mathrm{obs}} = (\boldsymbol{\Sigma}_{k,\omega_n}^{-1} + \mathbf{W}_n^{\mathrm{obs}\,T} \boldsymbol{\Psi}_n^{\mathrm{obs}\,-1} \mathbf{W}_n^{\mathrm{obs}})^{-1}$.

Equation (58) is maximized by using M-steps:

$$
\pi_{k,\omega} \quad = \quad \frac{\sum_n \delta_{\omega \omega_n} \gamma_{nk}}{\sum_{n'} \delta_{\omega \omega_{n'}}} \quad (63)
$$

$$
\boldsymbol{\mu}_{k,\omega} \quad = \quad \frac{\sum_n \delta_{\omega \omega_n} \gamma_{nk} \langle \mathbf{s}_{n,\omega_n} \rangle}{\sum_{n'} \delta_{\omega \omega_{n'}} \gamma_{n'k}} \quad (64)
$$

$$
\boldsymbol{\Sigma}_{k,\omega} \quad = \quad \frac{\sum_n \delta_{\omega \omega_n} \gamma_{nk} (\langle \mathbf{s}_{n,\omega_n} \mathbf{s}_{n,\omega_n}^T \rangle - \boldsymbol{\mu}_k \langle \mathbf{s}_{n,\omega_n} \rangle^T - \langle \mathbf{s}_{n,\omega_n} \rangle \boldsymbol{\mu}_k^T + \boldsymbol{\mu}_k \boldsymbol{\mu}_k^T)}{\sum_{n'} \delta_{\omega \omega_{n'}} \gamma_{n'k}} \quad (65)
$$

$$
\mathbf{w}_r \quad = \quad \left[ \sum_n \lambda_{n,r} \boldsymbol{\Gamma}_{n,r}^{\mathrm{obj}} \mathbf{x}_{n,\omega_n}^{\mathrm{obj}} \langle \mathbf{s}_{n,\omega_n} \rangle^T \right] \left[ \sum_n \lambda_{n,r} \langle \mathbf{s}_{n,\omega_n} \mathbf{s}_{n,\omega_n}^T \rangle \right]^{-1} \quad (66)
$$

$$
\boldsymbol{\Psi}_{\mathrm{o}} \quad = \quad \mathrm{tr}_D \left[ \sum_{n=1}^{N} \boldsymbol{\Gamma}_n^{\mathrm{obj}} \boldsymbol{\Delta}_n \boldsymbol{\Gamma}_n^{\mathrm{obj}\,T} \right] \mathrm{tr}_D \left[ \sum_{n=1}^{N} \boldsymbol{\Gamma}_n^{\mathrm{obj}} \boldsymbol{\Gamma}_n^{\mathrm{obj}\,T} \right]^{-1} \quad (67)
$$

where $\mathbf{w}_r$ and $\boldsymbol{\Gamma}_{n,r}^{\mathrm{obs}}$ are the $r$-th rows of $\mathbf{W}$ and $\boldsymbol{\Gamma}_n^{\mathrm{obs}}$, respectively, and

$$
\boldsymbol{\Lambda}_n \quad \equiv \quad \boldsymbol{\Gamma}_n^{\mathrm{obs}} \boldsymbol{\Gamma}_n^{\mathrm{obs}\,T} = \mathrm{diag}(\lambda_{n,1}, \cdots, \lambda_{n,r}, \cdots) \quad (68)
$$

$$
\begin{aligned}
\boldsymbol{\Delta}_n \quad \equiv & \quad \mathbf{x}_{n,\omega_n}^{\mathrm{obj}} \mathbf{x}_{n,\omega_n}^{\mathrm{obj}\,T} - \mathbf{W}_n^{\mathrm{obs}} \langle \mathbf{s}_{n,\omega_n} \rangle \mathbf{x}_{n,\omega_n}^{\mathrm{obj}\,T} - \mathbf{x}_{n,\omega_n}^{\mathrm{obj}} (\mathbf{W}_n^{\mathrm{obs}} \langle \mathbf{s}_{n,\omega_n} \rangle)^T \\
& + \mathbf{W}_n^{\mathrm{obs}} \langle \mathbf{s}_{n,\omega_n} \mathbf{s}_{n,\omega_n}^T \rangle \mathbf{W}_n^{\mathrm{obs}\,T}. \quad (69)
\end{aligned}
$$

The posterior distribution of $\mathbf{x}^{\text{mis}}$ is given by

$$
\begin{aligned}
p(\mathbf{x}^{\text{mis}}|\mathbf{x}^{\text{obs}}) &= \int |d\mathbf{s}| p(\mathbf{x}^{\text{mis}}, \mathbf{s}|\mathbf{x}^{\text{obs}}) = \int |d\mathbf{s}| p(\mathbf{x}^{\text{mis}}|\mathbf{x}^{\text{obs}}, \mathbf{s}) p(\mathbf{s}|\mathbf{x}^{\text{obs}}) \\
&= \int |d\mathbf{s}| p(\mathbf{x}^{\text{mis}}|\mathbf{s}) p(\mathbf{s}|\mathbf{x}^{\text{obs}}) = \int |d\mathbf{s}| p(\mathbf{x}^{\text{mis}}|\mathbf{s}) \sum_k p(k|\mathbf{x}^{\text{obs}}) p(\mathbf{s}|\mathbf{x}^{\text{obs}}, k) \\
&= \sum_{k=1}^{K} p(k|\mathbf{x}^{\text{obs}}) \cdot |2\pi \mathbf{Z}_k^{\text{mis}}|^{-\frac{1}{2}} \exp\Big\{ -\frac{1}{2} (\mathbf{x}^{\text{mix}} - \langle \mathbf{x}^{\text{mix}} \rangle_k)^T \mathbf{Z}_k^{\text{mis}\,-1} \\
&\qquad\qquad\qquad\qquad\qquad\qquad (\mathbf{x}^{\text{mix}} - \langle \mathbf{x}^{\text{mix}} \rangle_k) \Big\}
\end{aligned}
\tag{70}
$$

where

$$
\mathbf{M}_{k,\text{red}}^{\text{obs}} \equiv (\mathbf{W}^{\text{mis}\,T} \boldsymbol{\Psi}^{\text{mis}\,-1} \mathbf{W}^{\text{mis}} + \mathbf{M}_k^{\text{obs}\,-1})^{-1}
\tag{71}
$$

$$
\mathbf{Z}_k^{\text{mis}} = \boldsymbol{\Psi}^{\text{mis}} \left( \boldsymbol{\Psi}^{\text{mis}} - \mathbf{W}^{\text{mis}} \mathbf{M}_{k,\text{red}}^{\text{obs}} \mathbf{W}^{\text{mis}\,T} \right)^{-1} \boldsymbol{\Psi}^{\text{mis}}
\tag{72}
$$

$$
\langle \mathbf{x}^{\text{mix}} \rangle_k = \mathbf{Z}_k^{\text{mis}} \boldsymbol{\Psi}^{\text{mis}\,-1} \mathbf{W}^{\text{mis}} \mathbf{M}_{k,\text{red}}^{\text{obs}} \mathbf{M}_k^{\text{obs}\,-1} \langle \mathbf{s} \rangle_k .
\tag{73}
$$

We can also estimate the missing data by

$$
\langle \mathbf{x}_{n,\omega_n}^{\text{mis}} \rangle_k = \mathbf{Z}_{k,\omega_n}^{\text{mis}} \boldsymbol{\Psi}^{\text{mis}\,-1} \mathbf{W}^{\text{mis}} \mathbf{M}_{k,\text{red}}^{\text{obs}} \mathbf{M}_k^{\text{obs}\,-1} \langle \mathbf{s}_{n,\omega_n} \rangle_k
\tag{74}
$$

$$
\langle \mathbf{x}_{n,\omega_n}^{\text{mis}} \rangle = \sum_{k=1}^{K_{\omega_n}} \gamma_{nk} \langle \mathbf{x}_{n,\omega_n}^{\text{mis}} \rangle_k
\tag{75}
$$

$$
\begin{aligned}
\langle \mathbf{x}_{n,\omega_n}^{\text{mis}} \mathbf{x}_{n,\omega_n}^{\text{mis}\,T} \rangle &= \sum_{k=1}^{K_{\omega_n}} \gamma_{nk} \left( \mathbf{Z}_{k,\omega_n}^{\text{mis}} + (\langle \mathbf{x}_{n,\omega_n}^{\text{mis}} \rangle_k - \langle \mathbf{x}_{n,\omega_n}^{\text{mis}} \rangle)(\langle \mathbf{x}_{n,\omega_n}^{\text{mis}} \rangle_k - \langle \mathbf{x}_{n,\omega_n}^{\text{mis}} \rangle)^T \right) \\
&\quad + \langle \mathbf{x}_{n,\omega_n}^{\text{mis}} \rangle \langle \mathbf{x}_{n,\omega_n}^{\text{mis}} \rangle^T .
\end{aligned}
\tag{76}
$$

## B    SUPPLEMENTARY RESULTS AND DISCUSSION

**Supplementary Results**.

Table 1 is a reproduced one of Table 2 of the SyncTwin paper. Table 1 itself shows our state-of-art results across different biases and data sizes. The training/testing dataset used was added by independent white noise $\sim \mathcal{N}(0, 0.1)$. Our results of mean absolute error are at the level of 0.07, which means the ideal performance that cannot be algorithmically more improved when we notice

| (numpy.abs(np.random.randn(10000000) * 0.1)).mean() = 0.07979. |
| --- |

Tables 3 and 2 are also supporting that our method is robust to data missingness and noises. As requested by the Reviewer, we performed more experiments (including the suggested MC-NMM and CGP) under diverse treatment times and over prolonged post-treatment durations as follows:

We also tested algorithms for incomplete data of $50\%$ missingness, and Table 3 shows that our method is robust to data missingness. SyncTwin's performance gradually decreases by the missingness rate and the training data size, whereas SSA-GMM keeps its performance (compare $N_0 = 1000$ results of Table 1 and Table 3) and drops at around $N_0 = 200$. The EM algorithms for incomplete data are derived in Appendix A.

Table 3: Mean absolute errors by SSA-GMM, SyncTwin, and SC on ITE for incomplete data of 50% missingness. The best performer is in bold.

| Method | $N = N_0 + N_1 = 200 + 200 = 400$ | | | $N = N_0 + N_1 = 1000 + 200 = 1200$ | | |
| --- | --- | --- | --- | --- | --- | --- |
| | $p_0 = 0.1$ | $p_0 = 0.25$ | $p_0 = 0.5$ | $p_0 = 0.1$ | $p_0 = 0.25$ | $p_0 = 0.5$ |
| SSA-GMM | **0.327**(.028) | 0.232 (.016) | **0.141**(.009) | **0.091**(.006) | **0.054**(.004) | **0.061**(.004) |
| SyncTwin | 0.351 (.038) | **0.180**(.014) | 0.148 (.011) | 0.202 (.018) | 0.142 (.010) | 0.113 (.007) |
| SC | 0.441 (.044) | 0.272 (.024) | 0.252 (.030) | 0.424 (.041) | 0.298 (.028) | 0.228 (0.021) |

Table 4: Mean absolute errors (MAE) on ITE under different lengths of the temporal covariates (or intervention times) $T_0$ and data sizes $N_0$ with no missing data and $p_0 = 0.5$ confounding bias level. Estimated standard deviations are shown in parentheses. The best performer is in bold.

| Method | $N = N_0 + N_1 = 200 + 200 = 400$ | | | $N = N_0 + N_1 = 1000 + 200 = 1200$ | | |
| --- | --- | --- | --- | --- | --- | --- |
| | $T_0 = 15$ $T = 20$ | $T_0 = 25$ $T = 30$ | $T_0 = 45$ $T = 50$ | $T_0 = 15$ $T = 20$ | $T_0 = 25$ $T = 30$ | $T_0 = 45$ $T = 50$ |
| SSA-GMM | **0.092**(.005) | **0.076**(.004) | **0.081**(.004) | **0.065**(.003) | **0.069**(.004) | **0.081**(.004) |
| SyncTwin | 0.121 (.009) | 0.128 (.008) | 0.120 (.007) | 0.097 (.005) | 0.094 (.005) | 0.085 (.004) |
| SC | 0.140 (.019) | 0.149 (.018) | 0.138 (.021) | 0.190 (.029) | 0.214 (.036) | 0.215 (.044) |
| RSC | 0.348 (.023) | 0.322 (.019) | 0.228 (.011) | * | 0.302 (.014) | * |
| MC-NNM | 0.454 (.023) | 0.226 (.011) | 0.159 (.008) | 0.140 (.007) | 0.124 (.006) | 0.109 (.005) |
| CFRNet | 0.316 (.025) | 0.291 (.003) | 0.143 (.008) | 0.353 (.035) | 0.104 (.007) | 0.095 (.005) |
| CRN | 0.307 (.022) | 0.335 (.023) | 0.316 (.022) | 0.282 (.018) | 0.563 (.035) | 0.457 (.028) |
| RMSN | 0.311 (.028) | 0.334 (.027) | 0.493 (.032) | 0.342 (.032) | 0.390 (.032) | 0.557 (.036) |
| CGP | 0.561 (.036) | 0.561 (.035) | 0.549 (.035) | 0.578 (.037) | 0.602 (.038) | 0.611 (.038) |
| 1NN | 1.356 (.072) | 1.614 (.078) | 1.575 (.078) | 1.322 (.072) | 1.384 (.083) | 1.744 (.098) |

**Pretreatment Match**. In the original synthetic control method and its variant models (e.g., robust synthetic control, SyncTwin, etc.), a synthetic control is defined as a weighted average of the units in the donor pool (control group). The post-treatment values of the weighted average are regarded as the counterfactual prediction of post-treatment outcomes. However, the pre-treatment values of the weighted average are not the same as the observed pre-treatment data because the problem of finding the optimal (nonnegative) weights is basically an over-determined linear system of equations or a linear system of equations that has no unique solution due to nonnegativity regularization, and it only finds an approximated solution.

Empirically, Figure 3 shows that there are gaps between the black line (factual data) and the blue line (SCM prediction) in the pre-treatment periods, which suggests the infringement that the pre-treatment

Table 5: Mean absolute errors by SSA-GMM, SyncTwin, and SC on ITE over prolonged post-treatment durations with no missing data and $p_0 = 0.5$ confounding bias level. The best performer is in bold.

| Method | $N = N_0 + N_1 = 200 + 200 = 400$ | | | $N = N_0 + N_1 = 1000 + 200 = 1200$ | | |
|---|---|---|---|---|---|---|
| | $T_0 = 25$ $T = 30$ | $T_0 = 25$ $T = 35$ | $T_0 = 25$ $T = 40$ | $T_0 = 25$ $T = 30$ | $T_0 = 25$ $T = 35$ | $T_0 = 25$ $T = 40$ |
| SSA-GMM | **0.076**(.004) | **0.094**(.005) | **0.122**(.005) | **0.069**(.004) | **0.056**(.003) | **0.045**(.002) |
| SyncTwin | 0.120 (.008) | 0.118 (.009) | 0.125 (.009) | 0.089 (.005) | 0.093 (.005) | 0.092 (.005) |
| SC | 0.150 (.018) | 0.189 (.032) | 0.192 (.032) | 0.158 (.023) | 0.142 (.020) | 0.222 (.037) |

predictions (blue line) are also affected reversely by the policy in 1988. Furthermore, Figure 4 of the SyncTwin paper also shows the infringement. The authors say the synthetic twin matches its pre-treatment outcomes "closely," but our SSA-GMM results "exactly" match any pre-treatment outcomes. Figure 2 of the CGP paper (Schulam & Saria, 2017) also shows the gaps between the observed pre-treatment measurements (green points) and the model prediction (blue curves). Even a very small difference in the pre-treatment periods usually can make a big difference in the post-treatment prediction, and the restriction cannot be ignored. Our contribution to the self-evident constraint is clear and not limited.

**EM algorithm**. When the number of mixture components is $K = 1$, we can ensure the unique solution of target parameters under rotational and scaling ambiguity (i.e., $\mathbf{WS} = (\mathbf{WR})(\mathbf{R}^{-1}\mathbf{S}) = \mathbf{W}'\mathbf{S}'$) that gives a global maximization of the likelihood function. However, a Gaussian mixture model with $K > 1$ is not free of local maxima issues, and EM algorithms only find local optimal solutions of the likelihood function. Our model also does not give the same unique parameters every time it is trained. Normally, however, as we are not too concerned about non-unique solutions from the conventional use of a Gaussian mixture model for density estimation, the authors do not think it is a concerning issue for the generative task. Empirically, we observe that the probability of getting some worse solutions is very low.

**Linear Observers**. Our method is a unique algorithm distinguished from the SCM-based framework. SCM (synthetic control method) algorithms do find a convex combination of the control group data to generate a synthetic control. Originally, SCM does not use linear observers like our $\mathbf{W}^{(t)}$ and does not use a latent representation of data like our state vector representation $\mathbf{S}$. It is true that SyncTwin uses a latent representation (as shown in Figure 2 of their paper), which makes it look similar to ours, but SyncTwin also finds a convex combination in the latent representation space (as shown by Equation 4 of their paper) like the SCM-based methods. Our method does not solve such a combinatorial problem to generate counterfactuals. Our method is a probabilistic generative model that can randomly generate counterfactuals from the null space (spanned by $\mathbf{W}_{\perp}^{(T_0)}$) to pretreatment observers, which ensures the pretreatment causality restriction.

**Additive intervention**. This paper emphasizes the restriction on the counterfactual generation that receiving treatment at a time cannot cause any difference reversely to pretreatment observation data. The restriction is obeyed in this paper by assuming such linear and additive intervention. Because the linear and additive intervention is made only in the null space to pretreatment observers (as shown in Assumption 4 and Definition 2.2), the intervention does not cause any effect on pretreatment observation data. If we decided to use nonlinear observers, we would have to assume a nonlinear intervention to keep the restriction. In a bigger frame, we can consider two equally qualified options: one is that we assume a linear observer system and employ a Gaussian mixture model, and the other is that we assume a nonlinear observer system and use a single Gaussian model. This paper takes the first option and, in the first option, the assumption of linear and additive intervention is reasonable.

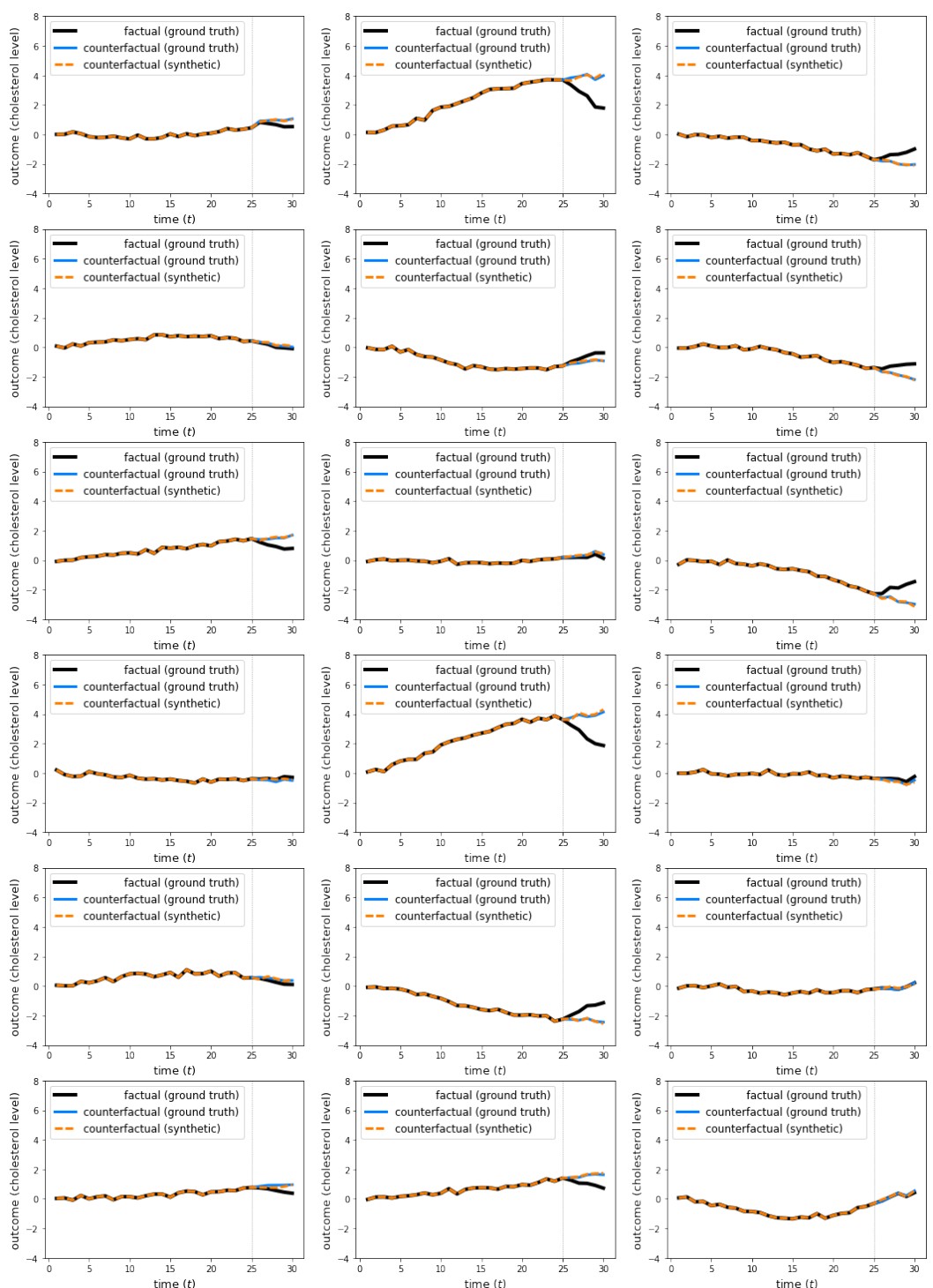

Figure 4: Results from a complete dataset of $N_0 = 200$ and $p_0 = 0.1$. We used parameters of $K_0 = 2$ and $M = 85$. All the factual data are given from the statins treatment group.

## C    PROOFS

*Proof of Lemma 2.1.* The cross-sectional probability density function is given as

$$
\begin{aligned}
\frac{\mathcal{N}(\mathbf{s} \in \mathcal{P}_0 | \boldsymbol{\mu}, \boldsymbol{\Sigma})}{\int_{\mathbf{s} \in \mathcal{P}_0} \mathcal{N}(\mathbf{s} | \boldsymbol{\mu}, \boldsymbol{\Sigma})} &= \frac{\mathcal{N}(\mathbf{W}_0 \boldsymbol{\xi} + \mathbf{s}_0 | \boldsymbol{\mu}, \boldsymbol{\Sigma})}{\int_{\boldsymbol{\xi}} \mathcal{N}(\mathbf{W}_0 \boldsymbol{\xi} + \mathbf{s}_0 | \boldsymbol{\mu}, \boldsymbol{\Sigma})} \\
&= \frac{\mathcal{N}(\mathbf{W}_0 \boldsymbol{\xi} | \boldsymbol{\mu} - \mathbf{s}_0, \boldsymbol{\Sigma})}{\mathcal{N}(\mathbf{0} | \boldsymbol{\mu} - \mathbf{s}_0, \boldsymbol{\Sigma}) \,/\, \mathcal{N}(\mathbf{0} | \mathbf{W}_0^T \boldsymbol{\Sigma}^{-1}(\boldsymbol{\mu} - \mathbf{s}_0), \mathbf{W}_0^T \boldsymbol{\Sigma}^{-1} \mathbf{W}_0)} \\
&= \mathcal{N}\left(\boldsymbol{\xi} \,|\, (\mathbf{W}_0^T \boldsymbol{\Sigma}^{-1} \mathbf{W}_0)^{-1} \mathbf{W}_0^T \boldsymbol{\Sigma}^{-1}(\boldsymbol{\mu} - \mathbf{s}_0), \, (\mathbf{W}_0^T \boldsymbol{\Sigma}^{-1} \mathbf{W}_0)^{-1}\right).
\end{aligned}
$$

$\square$

*Proof of Lemma 2.2.* The cross-sectional Gaussian mixture distribution is given as

$$
\frac{\sum_k \pi_k \mathcal{N}(\mathbf{s} \in \mathcal{P}_0 | \boldsymbol{\mu}_k, \boldsymbol{\Sigma}_k)}{\int_{\mathbf{s} \in \mathcal{P}_0} \sum_{k'} \pi_{k'} \mathcal{N}(\mathbf{s} | \boldsymbol{\mu}_{k'}, \boldsymbol{\Sigma}_{k'})} = \frac{\sum_k \pi_k \mathcal{N}(\mathbf{W}_0 \boldsymbol{\xi} + \mathbf{s}_0 | \boldsymbol{\mu}_k, \boldsymbol{\Sigma}_k)}{\int_{\boldsymbol{\xi}} \sum_{k'} \pi_{k'} \mathcal{N}(\mathbf{W}_0 \boldsymbol{\xi} + \mathbf{s}_0 | \boldsymbol{\mu}_{k'}, \boldsymbol{\Sigma}_{k'})}
$$

$$
= \frac{\sum_k \pi_k \mathcal{N}(\mathbf{W}_0 \boldsymbol{\xi} | \boldsymbol{\mu}_k - \mathbf{s}_0, \boldsymbol{\Sigma}_k)}{\sum_{k'} \pi_{k'} \mathcal{N}(\mathbf{0} | \boldsymbol{\mu}_{k'} - \mathbf{s}_0, \boldsymbol{\Sigma}_{k'}) \,/\, \mathcal{N}(\mathbf{0} | \mathbf{W}_0^T \boldsymbol{\Sigma}_{k'}^{-1}(\boldsymbol{\mu}_{k'} - \mathbf{s}_0), \mathbf{W}_0^T \boldsymbol{\Sigma}_{k'}^{-1} \mathbf{W}_0)}
$$

$$
= \sum_k \frac{\pi_k \mathcal{N}(\mathbf{0} | \boldsymbol{\mu}_k - \mathbf{s}_0, \boldsymbol{\Sigma}_k) \,/\, \mathcal{N}(\mathbf{0} | \mathbf{W}_0^T \boldsymbol{\Sigma}_k^{-1}(\boldsymbol{\mu}_k - \mathbf{s}_0), \mathbf{W}_0^T \boldsymbol{\Sigma}_k^{-1} \mathbf{W}_0)}{\sum_{k'} \pi_{k'} \mathcal{N}(\mathbf{0} | \boldsymbol{\mu}_{k'} - \mathbf{s}_0, \boldsymbol{\Sigma}_{k'}) \,/\, \mathcal{N}(\mathbf{0} | \mathbf{W}_0^T \boldsymbol{\Sigma}_{k'}^{-1}(\boldsymbol{\mu}_{k'} - \mathbf{s}_0), \mathbf{W}_0^T \boldsymbol{\Sigma}_{k'}^{-1} \mathbf{W}_0)} \times
$$

$$
\frac{\mathcal{N}(\mathbf{W}_0 \boldsymbol{\xi} | \boldsymbol{\mu}_k - \mathbf{s}_0, \boldsymbol{\Sigma}_k)}{\pi_k \mathcal{N}(\mathbf{0} | \boldsymbol{\mu}_k - \mathbf{s}_0, \boldsymbol{\Sigma}_k) \,/\, \mathcal{N}(\mathbf{0} | \mathbf{W}_0^T \boldsymbol{\Sigma}_k^{-1}(\boldsymbol{\mu}_k - \mathbf{s}_0), \mathbf{W}_0^T \boldsymbol{\Sigma}_k^{-1} \mathbf{W}_0)}
$$

$$
= \sum_k \frac{\pi_k \mathcal{N}(\mathbf{W}_0 \mathbf{m}_k + \mathbf{s}_0 | \boldsymbol{\mu}_k, \boldsymbol{\Sigma}_k) |\mathbf{C}_k|^{\frac{1}{2}}}{\sum_{k'} \pi_{k'} \mathcal{N}(\mathbf{W}_0 \mathbf{m}_{k'} + \mathbf{s}_0 | \boldsymbol{\mu}_{k'}, \boldsymbol{\Sigma}_{k'}) |\mathbf{C}_{k'}|^{\frac{1}{2}}} \mathcal{N}(\boldsymbol{\xi} | \mathbf{m}_k, \mathbf{C}_k)
$$

where $\mathbf{m}_k = (\mathbf{W}_0^T \boldsymbol{\Sigma}_k^{-1} \mathbf{W}_0)^{-1} \mathbf{W}_0^T \boldsymbol{\Sigma}_k^{-1}(\boldsymbol{\mu}_k - \mathbf{s}_0)$ and $\mathbf{C}_k = (\mathbf{W}_0^T \boldsymbol{\Sigma}_k^{-1} \mathbf{W}_0)^{-1}$ by applying Lemma 2.1 to each Gaussian. $\square$

*Proof of Theorem 2.3.* From Definitions 2.2 and 4, we have for the $n$-th unit with treatment $\omega_n$

$$
\begin{aligned}
\mathbf{x}_{n,\bar{\omega}_n}^{(t)} - \mathbf{x}_{n,\omega_n}^{(t)} &= \mathbf{W}^{(t)}(\mathbf{s}_{n,\bar{\omega}_n} - \langle \mathbf{s}_{n,\omega_n} \rangle) \\
&= \mathbf{W}^{(t)} \mathbf{N}^{(T_0)} \boldsymbol{\xi}_{n,\bar{\omega}_n}
\end{aligned}
$$

where $\boldsymbol{\xi}_{n,\bar{\omega}_n}$ is given by Lemma 2.2 at $\mathbf{W}_0 = \mathbf{N}^{(T_0)}$, $\pi_k = \pi_{k,\bar{\omega}_n}$, $\boldsymbol{\mu}_k = \boldsymbol{\mu}_{k,\bar{\omega}_n}$, $\boldsymbol{\Sigma}_k = \boldsymbol{\Sigma}_{k,\bar{\omega}_n}$, $\mathbf{s}_0 = \langle \mathbf{s}_{n,\omega_n} \rangle$. $\square$

*Proof of Corollary 2.3.1.* From Definitions 2.1, 2.2, and Assumption 2, we have

$$
\mathbf{x}_\omega^{(t)} = \mathbf{W}^{(t)} \mathbf{s}_\omega + \boldsymbol{\eta}^{(t)} = \mathbf{W}^{(t)}(\mathbf{s}_{\bar{\omega}} + \mathbf{N}^{(T_0)} \boldsymbol{\xi}) + \boldsymbol{\eta}^{(t)} = \mathbf{W}^{(t)} \mathbf{s}_{\bar{\omega}} + \boldsymbol{\eta}^{(t)} \equiv \mathbf{x}_{\bar{\omega}}^{(t)}
$$

where $t = 1, \cdots, T_0$. $\square$

*Proof of Corollary 2.3.2.* ITE is the difference of treated outcome from untreated outcome. From Definitions 2.2 and 4,

$$
\begin{aligned}
\text{ITE}^{(t)} &= \mathbf{w}_{\text{effect}}^T \left[ (2\omega - 1)\mathbf{W}^{(t)} \mathbf{s} + (1 - 2\omega)\mathbf{W}^{(t)}(\mathbf{s} + \mathbf{N}^{(T_0)} \boldsymbol{\xi}) \right] \\
&= (1 - 2\omega)\mathbf{w}_{\text{effect}}^T \mathbf{W}^{(t)} \mathbf{N}^{(T_0)} \boldsymbol{\xi}
\end{aligned}
$$

where $\boldsymbol{\xi}$ is a random variable from Lemma 2.2 by substituting $\mathbf{W}_0$ with $\mathbf{N}^{(T_0)}$ and $\mathbf{w}_{\text{effect}}$ is explained in the main paper. $\square$

*Proof of Theorem 2.4.* From Theorem 2.3.2 and Eq. (7),

$$
\begin{aligned}
\text{E}[\text{ITE}^{(t)}] &= (1 - 2\omega)\mathbf{w}_{\text{effect}}^T \mathbf{W}^{(t)} \mathbf{N}^{(T_0)} \text{E}[\boldsymbol{\xi}] \\
&= (1 - 2\omega)\mathbf{w}_{\text{effect}}^T \mathbf{W}^{(t)} \mathbf{N}^{(T_0)} \cdot \sum_k p_{\bar{\omega}k} \mathbf{m}_{\bar{\omega}k},
\end{aligned}
$$

$$
\begin{aligned}
\text{Cov}[\boldsymbol{\xi}] &= \text{E}[\boldsymbol{\xi}\boldsymbol{\xi}^T] - \text{E}[\boldsymbol{\xi}]\text{E}[\boldsymbol{\xi}]^T \\
&= \sum_k p_{\bar{\omega}k}(\mathbf{C}_{\bar{\omega}k} + \mathbf{m}_{\bar{\omega}k}\mathbf{m}_{\bar{\omega}k}^T) - (\sum_k p_{\bar{\omega}k}\mathbf{m}_{\bar{\omega}k})(\sum_k p_{\bar{\omega}k}\mathbf{m}_{\bar{\omega}k})^T \\
&= \sum_k p_{\bar{\omega}k}(\mathbf{C}_{\bar{\omega}k} + (\mathbf{m}_{\bar{\omega}k} - \sum_{k'} p_{\bar{\omega}k'}\mathbf{m}_{\bar{\omega}k'})(\mathbf{m}_{\bar{\omega}k} - \sum_{k'} p_{\bar{\omega}k'}\mathbf{m}_{\bar{\omega}k'})^T) \\
&\equiv \sum_k p_{\bar{\omega}k}(\mathbf{C}_{\bar{\omega}k}^{\text{within}} + \mathbf{C}_{\bar{\omega}k}^{\text{between}}),
\end{aligned}
$$

and

$$
\begin{aligned}
&\text{E}[(\text{ITE}^{(t)} - \text{E}[\text{ITE}^{(t)}])^2] \\
&= (1 - 2\omega)^2 \mathbf{w}_{\text{effect}}^T \mathbf{W}^{(t)} \mathbf{N}^{(T_0)} \cdot \text{Cov}[\boldsymbol{\xi}] \cdot \mathbf{N}^{(T_0)T} \mathbf{W}^{(t)T} \mathbf{w}_{\text{effect}} \\
&= \mathbf{w}_{\text{effect}}^T \mathbf{W}^{(t)} \mathbf{N}^{(T_0)} \cdot \sum_k p_{\bar{\omega}k}(\mathbf{C}_{\bar{\omega}k}^{\text{within}} + \mathbf{C}_{\bar{\omega}k}^{\text{between}}) \cdot \mathbf{N}^{(T_0)T} \mathbf{W}^{(t)T} \mathbf{w}_{\text{effect}}.
\end{aligned}
$$

$\square$

