# OpenReview forum: "A Linear Algebraic Framework for Counterfactual Generation"
_ICLR.cc/2024/Conference — ICLR 2024 poster_

### Official Review · Reviewer_aGd4 · 2023-10-29

**Soundness:** 3 good
**Presentation:** 3 good
**Contribution:** 2 fair
**Rating:** 6
**Confidence:** 3

**Summary:**

This paper investigates the problem of generating synthetic counterfactual data that exactly matches pretreatment factual data under a linear algebraic framework (namely, the static state of the factual and counterfactual data are all in affine subspace). By defining the shift from a factual state to a counterfactual state under a soft intervention, the counterfactual data can be generated by solving the log-likelihood function as a static state analysis with priors of an ω-dependent Gaussian mixture model. Such a generation method can be used to estimate an ITE and the experimental results of the simulated data show that the proposed method outperforms the baselines in estimating ITE.

**Strengths:**

1. This paper deals with a challenging but important problem of counterfactual generation, which may be useful to the field of causal inference.

2. A practical estimating method is provided in estimating parameters (EM algorithms) and formalizes a generation process of counterfactual data according to estimated results.

3. The experimental results verify the generated counterfactual data is useful for estimating ITE.

**Weaknesses:**

1. My main concern is the identification of parameters in the log-likelihood function, i.e., how to ensure the unique solution of target parameters by the EM algorithm. Because this is a key issue in generating the true counterfactual data.

2. I think it is necessary to discuss the reasonableness for the same linear observation "W" in the factual data and counterfactual data generation. Because if we resolve the "W" from factual data, we can naturally obtain the counterfactual state under this setting. Such a method does not surprise me and I guess there are some similarities with the SCM-based framework (used for estimating counterfactual framework).

3. Assumption 4 assumes that the soft intervertion on "s" is linearity and additive. Some limitations or reasonableness to it also require to be discussed.

Besides, as this is a bit out of my expertise, my confidence on the score is on the lower spectrum.

**Questions:**

Refer to "Weaknesses".

---

> ### Author Response · Authors · 2023-11-17
>
> Thanks for the careful review that brought out several subtle observations that need clarification. We provide a perspective. We will ensure that we include these to improve the paper.
>
> 1.	EM algorithm.  When the number of mixture components is $K=1$, we can ensure the unique solution of target parameters under rotational and scaling ambiguity (i.e., $\mathbf{WS} = (\mathbf{WR})(\mathbf{R}^{-1}\mathbf{S}) = \mathbf{W'S'}$) that gives a global maximization of the likelihood function. However, a Gaussian mixture model with $K > 1$ is not free of local maxima issues, and EM algorithms only find local optimal solutions of the likelihood function. Our model also does not give the same unique parameters every time it is trained. Normally, however, as we are not too concerned about non-unique solutions from the conventional use of a Gaussian mixture model for density estimation, the authors do not think it is a concerning issue for the generative task. Empirically, we observe that the probability of getting some worse solutions is very low.
>
> 2.	Linear Observers. Our method is a unique algorithm distinguished from the SCM-based framework. SCM (synthetic control method) algorithms do find a convex combination of the control group data to generate a synthetic control. Originally, SCM does not use linear observers like our $\mathbf{W}^{(t)}$ and does not use a latent representation of data like our state vector representation $\mathbf{S}$. It is true that SyncTwin uses a latent representation (as shown in Figure 2 of their paper), which makes it look similar to ours, but SyncTwin also finds a convex combination in the latent representation space (as shown by Equation 4 of their paper) like the SCM-based methods. Our method does not solve such a combinatorial problem to generate counterfactuals. Our method is a probabilistic generative model that can randomly generate counterfactuals from the null space (spanned by $\mathbf{W}_\perp^{(T_0)}$) to pretreatment observers, which ensures the pretreatment causality restriction.
>
> 3.	Linear and additive intervention. This paper emphasizes the restriction on the counterfactual generation that receiving treatment at a time cannot cause any difference reversely to pretreatment observation data. The restriction is obeyed in this paper by assuming such linear and additive intervention. Because the linear and additive intervention is made only in the null space to pretreatment observers (as shown in Assumption 4 and Definition 2.2), the intervention does not cause any effect on pretreatment observation data. If we decided to use nonlinear observers, we would have to assume a nonlinear intervention to keep the restriction. In a bigger frame, we can consider two equally qualified options: one is that we assume a linear observer system and employ a Gaussian mixture model and the other is that we assume a nonlinear observer system and use a single Gaussian model. This paper takes the first option and, in the first option, the assumption of linear and additive intervention is reasonable.

---

### Official Review · Reviewer_TJyJ · 2023-11-01

**Soundness:** 3 good
**Presentation:** 3 good
**Contribution:** 2 fair
**Rating:** 6
**Confidence:** 3

**Summary:**

The paper proposes a linear algebraic framework for generating synthetic counterfactual data that matches pretreatment factual data. The approach claims to be the first-ever counterfactual generative model for creating personalized clinical trial digital twins. The method outperforms other methods of counterfactual generation and individual treatment effect estimation, as demonstrated using simulated ground truth counterfactual data. Additionally, the paper provides a formula to estimate the time-varying variance of individual treatment effects, which can be interpreted as a confidence measure for the generated counterfactuals.

**Strengths:**

- The paper is well-written and easy to follow.

- The experiments on California cigarette sales data and simulated LDL cholesterol data showcase the effectiveness of the proposed method as some extent.

**Weaknesses:**

- The motivation of this paper is unclear. The contribution also seems limited. The restriction on counterfactual generation that receiving treatment at a time $T_0$ cannot cause any difference reversely to counterfactual generation at $t < T_0$, comes across as self-evident. It remains ambiguous whether prior research infringed upon this constraint. The comparative advantages of the introduced method with the existing work are not distinctly highlighted, leading to a gap in understanding its unique value proposition.

- To evaluate how true synthetic counterfactuals are, more experiments on synthetic datasets including ground-truth counterfactual data are required. It would be insightful to demonstrate the method's robustness under diverse treatment times and over prolonged post-treatment durations.

- The paper could benefit from a rigorous theoretical exploration addressing the method's resilience against noise and confounding bias. Would this be challenging with such a linear framework?

- There is a lack of literature review or an introduction to some of the methods appearing in the experimental part, such as MC-NNM (Athey et al., 2021) and CGP (Schulam and Saria, 2017).

**Questions:**

Regarding Fig.4, what methodology did the authors employ to randomly produce counterfactual pre-treatment data?

---

> ### Author Response · Authors · 2023-11-18
>
> Thanks for the suggestions on clarifying the assumptions, comparative advantages, and more rigorous theoretical treatment. We provide brief explanations, results, and plans for rigorous treatment.
>
> (1)  In the original synthetic control method and its variant models (e.g., robust synthetic control, SyncTwin, etc.), a synthetic control is defined as a weighted average of the units in the donor pool (control group). The post-treatment values of the weighted average are regarded as the counterfactual prediction of post-treatment outcomes. However, the pre-treatment values of the weighted average are not the same as the observed pretreatment data because the problem of finding the optimal (nonnegative) weights is basically an over-determined linear system of equations or a linear system of equations that has no unique solution due to nonnegativity regularization, and it only finds an approximated solution.
>
> Empirically, Figure 3 of our paper shows that there are gaps between the black line (factual data) and the blue line (SCM prediction) in the pretreatment periods, which suggests the infringement that the pretreatment predictions (blue line) are also affected reversely by the policy in 1988.
> Furthermore, Figure 4 of the SyncTwin paper also shows the infringement. The authors say:
>
> *“The individual shown in Figure 4 (top) has a sensible ITE estimate because the synthetic twin matches its pretreatment outcomes closely. In addition to visualization, we can calculate the individualized error bound based on $d_i^y$ (Equation 7).”*
>
> They say the synthetic twin matches its pretreatment outcomes "closely", but our SSA-GMM results “exactly” match any pretreatment outcomes. Figure 2 of the CGP paper (Schulam \& Saria, 2017) also shows the gaps between the observed pretreatment measurements (green points) and the model prediction (blue curves). Even a very small difference in the pretreatment periods can make a big difference in the post-treatment prediction, and the restriction cannot be ignored. Our contribution to the self-evident constraint is clear and not limited.
>
> (2) Table 1 in our paper is a reproduced one of Table 2 of the SyncTwin paper. Table 1 itself shows our state-of-the-art results across different biases and data sizes. The training/testing dataset used was added by independent white noise $\sim\mathcal{N}(0, 0.1)$. Our results of a mean absolute error are at the level of 0.07, which means the ideal performance that cannot be algorithmically further improved when we notice
>
>       (numpy.abs(numpy.random.randn(10000000) * 0.1)).mean() = 0.07979.
>
> Tables 2 and 3 of our paper also support that our method is robust to data missingness and noises. As requested by the Reviewer, we performed more experiments under diverse treatment times and over prolonged post-treatment durations. Our new tables in **Table A** and **Table B** show the state-of-the-art performance of our method.
>
> (3) This paper could benefit more from a rigorous theoretical exploration against noise, bias, and data missingness. This is a linear framework that enables it to be analyzed theoretically; we can include theoretical analyses in the Appendix and refer to it in the main body of the paper. In the appendix, we will also provide a literature review on those methods, such as MC-NNM and CGP.
>
> (4) To produce ground truth counterfactual pretreatment and post-treatment data in Figure 4, we used the same Pharmacology model [(link)](https://onlinelibrary.wiley.com/doi/abs/10.1111/j.1472-8206.2006.00404.x) and codes [(link)](https://github.com/ZhaozhiQIAN/SyncTwin-NeurIPS-2021/blob/main/sim/pkpd.py) that SyncTwin used.

---

> ### Author Response · Authors · 2023-11-18
>
> **Table A**: Performance comparison under diverse treatment times
>
>        Method            N = N_0 + N_1 = 200 + 200 = 400                N = N_0 + N_1 = 1000 + 200 = 1200
>
>                   T_0=15, T=20    T_0=25, T=30    T_0=45, T=50    T_0=15,  T=20    T_0=25, T=30    T_0=45, T=50
>
>       SSA-GMM     0.092 (.005)    0.076 (.004)    0.081 (.004)    0.065 (.003)     0.069 (.004)    0.081 (.004)
>
>       SyncTwin    0.121 (.009)    0.128 (.008)    0.120 (.007)    0.097 (.005)     0.094 (.005)    0.085 (.004)
>
>       SC          0.140 (.019)    0.149 (.018)    0.138 (.021)    0.190 (.029)     0.214 (.036)    0.215 (.044)
>
>       RSC         0.348 (.023)    0.322 (.019)    0.228 (.011)      *              0.302 (.014)       *
>
>       MC-NNM      0.454 (.023)    0.226 (.011)    0.159 (.008)    0.140 (.007)     0.124 (.006)    0.109 (.005)
>
>       CFRNet      0.316 (.025)    0.291 (.003)    0.143 (.008)    0.353 (.035)     0.104 (.007)    0.095 (.005)
>
>       CRN         0.307 (.022)    0.335 (.023)    0.316 (.022)    0.282 (.018)     0.563 (.035)    0.457 (.028)
>
>       RMSN        0.311 (.028)    0.334 (.027)    0.493 (.032)    0.342 (.032)     0.390 (.032)    0.557 (.036)
>
>       CGP         0.561 (.036)    0.561 (.035)    0.549 (.035)    0.578 (.037)     0.602 (.038)    0.611 (.038)
>
>
>   **Table B**: Performance comparison over prolonged post-treatment durations
>
>        Method            N = N_0 + N_1 = 200 + 200 = 400                N = N_0 + N_1 = 1000 + 200 = 1200
>
>                   T_0=25, T=30    T_0=25, T=35    T_0=25, T=40    T_0=25,  T=30    T_0=25, T=35    T_0=25, T=40
>
>       SSA-GMM     0.076 (.004)    0.094 (.005)    0.122 (.005)    0.069 (.004)     0.056 (.003)    0.045 (.002)
>
>       SyncTwin    0.120 (.008)    0.118 (.009)    0.125 (.009)    0.089 (.005)     0.093 (.005)    0.092 (.005)
>
>       SC          0.150 (.018)    0.189 (.032)    0.192 (.032)    0.158 (.023)     0.142 (.020)    0.222 (.037)
>
> where $T$ is the total length of periods starting from $t=1$, and $T_0$ is the intervention time.

---

> > ### Author Response · Authors · 2023-11-23
> > **To Reviewers**
> >
> > We sincerely provided additional information and results to resolve the weak points. Since we have uploaded the final version of the paper, we would appreciate it if you could finally comment on whether they have been sufficiently explained or are still lacking. Together with Appendix B, the final version describes why it is robust against bias through the newly updated analysis of Figure 4 to resolve any doubts reviewers may have despite presenting state-of-the-art results.

---

> > > ### Comment · Reviewer_TJyJ · 2023-11-23
> > > **Thanks.**
> > >
> > > I thank the authors for their careful responding to all the comments and the revision of the paper. I will raise my score.

---

### Official Review · Reviewer_EGeA · 2023-11-01

**Soundness:** 2 fair
**Presentation:** 2 fair
**Contribution:** 2 fair
**Rating:** 3
**Confidence:** 4

**Summary:**

This paper focuses on counterfactual generation. In particular, it introduces a Gaussian mixture model as a prior probability distribution over the static states, and uses it as a generative model for synthetic counterfactual longitudinal data. The proposed method can be used to estimate the individual treatment effects (ITE) using the synthetic counterfactual outcomes data.

**Strengths:**

This paper considers counterfactual generation, which is an important problem. Also, it uses a Mixture of Gaussians to approximate general data distributions; this is the main difference compared to previous papers.

**Weaknesses:**

The difference of the proposed method is that it assumes a Gaussian mixture model to approximate distributions, in contrast to previous works which employed neural networks. However, the experimental results may not provide sufficient evidence to fully assess the performance of the proposed method. Notably, the paper lacks experimental results on benchmark datasets such as IHDP and Jobs.

The paper asserts its ability to identify ITE as defined in Equation 11 and provides results on ITE estimation accuracy. However, it's important to note that the defined ITE is a random variable. Consequently, I'm uncertain whether the reported results pertain to ITE itself or its expected value.

**Questions:**

Are the reported results regarding ITE or the expected value of ITE?

---

> ### Author Response · Authors · 2023-11-17
>
> Thanks for your insights on the need to provide clarity on the main contribution and perceived weakness. We will definitely address this.
>
> Briefly,  our crucial insight and contribution is that our algorithm assumes the additivity between factual data and counterfactual predictions (by Assumption 4, Definition 2.2, and Definition 2.3) and that counterfactual predictions are made from in the null space (spanned by $\mathbf{W}_\perp^{(T_0)}$) to a set of the pretreatment linear observer matrices, which are referred to as a linear algebraic framework in the title. We used the Gaussian mixture model merely as a density estimation method to be able to derive analytic formulas of Equations (13) and (14) that are simple and practical. Such a model can be trained with a small dataset like clinical trial data.
>
> The Gaussian mixture model is replaceable by any other model as long as it can universally approximate the underlying data distributions. If it is replaced with a deep generative model, a new cross-sectional model is necessary and can be derived like Lemma 2.2.
>
> The model whose targeted problem and data are closest to ours is SyncTwin. The SyncTwin performance was demonstrated only by LDL cholesterol data, and we used the same dataset to show our state-of-the-art results. IHDP and JOBS datasets do not have time-varying longitudinal outcomes and are not in our target problems, so we did not consider them for comparison. We can compare performance with them with an appropriate reformulation of our algorithm in the Appendix if needed.
>
> Our results in Table 1 were produced using Equation (12) as a random variable because $\xi$ is a random variable, and $\text{ITE}_n^{(t)}$ is also a random variable. We repeated generating such random ITEs and numerically calculated their average and standard deviation, which are our results in Table 1. In Figure 3, on the contrary, the red solid line and pink shade are the expected values and standard deviations as calculated using the formulas of Equations (13) and (14).

---

### Official Review · Reviewer_njab · 2023-11-02

**Soundness:** 3 good
**Presentation:** 3 good
**Contribution:** 3 good
**Rating:** 6
**Confidence:** 3

**Summary:**

This work focuses on estimating time-varying individualized treatment effects within a linear model. The authors formulate a linear static-state model and provide a novel method to estimate the corresponding counterfactual data. They have demonstrated that the proposed approach can achieve better performance in their experiments.

**Strengths:**

1. Generating the counterfactual data is a challenging task.

2. The proposed method is both novel and interesting.

3. The analyses are sound and presented in a logical manner.

**Weaknesses:**

1. The method is only applicable to linear models.

2. In some instances, the description is not sufficiently clear. For example, the model's explanation here lacks clarity, particularly in the description of assumption 2, which necessitates further comprehension through reading the instructions in the appendix.

**Questions:**

Can you provide an intuitive explanation for Assumption 2? Typically, people use a linear dynamical system model to describe the generative mechanism. It would be beneficial to include some descriptions in the main text rather than placing them in the appendix.

---

> ### Author Response · Authors · 2023-11-17
>
> We appreciate the suggestion to improve our manuscript to bring out the contribution clearly. The main body of the paper presents a probabilistic model and Appendix D discusses its deterministic model as a matrix factorization problem without a prior distribution of Gaussian mixture. We will organize Appendix D so that readers can understand them clearly. It is page-limited to include Appendix D in the main text, and we will refer to it properly in the main text.
>
> In the real world, objects change with time, and so does the data. Transforming such time-varying data into a static time-invariant representation leads to a problem that is easier to solve, which is the motivation and foundation of our approach. In that light, Assumptions 1 and 2 provide a new perspective in which the objects’ new representation, denoted by $\mathbf{S}$, does not change, and the observer's viewpoint, denoted by $\mathbf{W}$, is changing. Such static representations of $\mathbf{S}$ that are invariant to time can be learned well with simpler models (such as in this paper, GMM). Assumption 2 describes the relationship between time-invariant data $\mathbf{S}$ and time-varying data $\mathbf{X}$. This observation is our main insight, which is then embellished by creating a more realistic model where one can include some additive measurement noise.  We further highlight that the validity of our insight and assumption is established by experimental results that offer a definite improvement over the state of the art.

---

### Meta-Review · Area_Chair_dtkw · 2023-12-06

**Metareview:**

The paper is concerned with counterfactual longitudinal data generation. This is done by assuming a Gaussian mixture model (or any other universal approximator of the underlying data distribution) as the prior over the static states. In any state, the result of an intervention is modeled as a random linear causal outcome. The reviewers had some issues about the clarity, such as the motivation and whether the reported results is the ITE itself or its expected value. The paper could benefit from a rewrite, but I found the results interesting enough to suggest accept if there is space.

**Justification For Why Not Higher Score:**

I found the work novel and interesting despite some of the valid concerns the reviewers raised so it there is room, I suggest we accept this paper.

**Justification For Why Not Lower Score:**

Counterfactual generation is an important topic and the work done by the authors is a step in the right direction.

---

### Decision · Program_Chairs · 2024-01-16

Accept (poster)